# Oxidation-driven acceleration of NPF-to-CCN conversion under polluted atmosphere: Evidence from mountain-top observations in Yangtze River Delta

Weibin Zhu[1,2], Sai Shang[1,2], Jieqi Wang[1,2], Yunfei Wu[1], Zhaoze Deng[1], Liang Ran[1], Ye Kuang[3], Guiqian Tang[1], Xiangpeng Huang[4], Xiaole Pan[1], Lanzhong Liu[5], Weiqi Xu[1], Yele Sun[1], Bo Hu[1], Zifa Wang[1], Zirui Liu*[1]

[1]State Key Laboratory of Atmospheric Environment and Extreme Meteorology, Institute of Atmospheric Physics, Chinese Academy of Sciences, Beijing, 100029, China

[2]University of Chinese Academy of Sciences, Beijing, 100049, China

[3]Institute for Environmental and Climate Research, Jinan University, Guangzhou, 511400, China

[4]College of Resources and Environmental Engineering, Jiangsu University of Technology, Changzhou, 213001, China

[5]Shanghuang Atmospheric Boundary Layer and Eco-Environment Observatory, Institute of Atmospheric Physics, Chinese Academy of Sciences, Jinhua, 321203, China

*Correspondence to*: Zirui Liu (liuzirui@mail.iap.ac.cn)

**Abstract.** To what extent the new particle formation (NPF) contributed to the cloud condensation nuclei (CCN) remained unclear, especially at the boundary layer top (BLT) in polluted atmosphere. Based on measurements at a mountain-top background site in southeastern China during spring 2024, this study systematically investigates the nucleation mechanism and subsequent growth dynamics of NPF events under contrasting air masses, and quantifies their role as a source of CCN. Eight NPF events were observed, and three of them occurred in the polluted conditions (NPF-P) which associated with regional transportation while the rest five events appeared in the clean conditions (NPF-C). The average formation rate ($J_{2.5}$: 2.4 vs. 0.7 $cm^{-3}$ $s^{-1}$) and growth rate (GR: 6.8 vs. 5.5 $nm$ $h^{-1}$) were significantly higher in NPF-P events than in NPF-C events, alongside elevated concentrations of sulfuric acid and ammonia. The correlation between log $J_3$ and [$H_2SO_4$], as well as theoretical simulations with the MALTE_BOX model, indicates that the enhanced nucleation in polluted conditions can be attributed to the participation of ammonia in stabilizing sulfuric acid-based clusters. In addition, much higher CCN enhancement factor was observed in NPF-P ($EF_{CCN}$: 1.6 vs. 0.7 in NPF-C) due to the regional transported of anthropogenic pollutants from the urban cluster regions and their secondary transformation under enhanced atmospheric oxidation capacity. Furthermore, the duration of NPF-to-CCN conversion was quantified using a 'Time Window ($\tau$)', revealing that polluted conditions accelerated the conversion by 17.0% ($\tau$ = 16.4 h vs. 19.8 h). Nitrate played an important role in maintaining a rapid particle growth rate, thereby shortening $\tau$ and enhancing CCN production from NPF—a process that can ultimately influence cloud microphysical properties by increasing the potential cloud droplet number concentration. These findings reveal that polluted air masses enhance both the efficiency and speed of CCN production at the BLT through elevated atmospheric oxidation capacity.

**Keywords:** New particle formation, Boundary layer top, Pollution condition, Cloud condensation nuclei, Particle growth, Atmospheric oxidation capacity.

## 1. Introduction

New Particle Formation (NPF) is the process in which low-volatility gaseous precursors nucleate to form stable nanoparticles, leading to rapid bursts in particle number concentration (Kulmala et al., 2001); these newly formed particles can subsequently grow to larger sizes via condensation of vapors or coagulation (Kerminen et al., 2018; Cai et al., 2024). As an important source of atmospheric particles, NPF profoundly influences cloud microphysical properties, radiative forcing, and precipitation efficiency through its conversion process to Cloud Condensation Nuclei (CCN), thereby regulating regional and even global climate systems (Laaksonen et al., 2005; Kalkavouras et al., 2017; Kalkavouras et al., 2019). Growth process of NPF events contributes to generating substantial CCN, with approximately half of CCN in the global troposphere potentially originating from NPF events (Zhao et al., 2024). Under polluted urban atmosphere, NPF event intensity enhanced, with growth processes potentially persisting for 2-3 days and leading to the formation of more particles capable of growing to CCN sizes (Zhang et al., 2019; Zhu et al., 2023). However, the contribution of NPF events to CCN exhibits considerable regional variability, and NPF may even suppress CCN activity under different conditions. However, high condensation sinks (CS) also resulting from higher background particle concentrations strongly suppress nanoparticle formation intensity, accelerate scavenging of small particles, and may reduce particle hygroscopicity, thereby diminishing contribution of NPF to CCN (Kalivitis et al., 2019). Consequently, while numerous long-term observational studies have established the general importance of NPF as a source of CCN, the specific chemical pathways governing particle formation and subsequent growth into CCN under varying atmospheric conditions, particularly at high-altitude sites influenced by complex pollution regimes, remain inadequately constrained and require further validation through targeted observations.

According to abundant field experiment observations, NPF typically manifests as "NPF events" within the global boundary layer; that is, the nucleation of nanoparticles and subsequent growth may occur over horizontal spatial scales extending up to tens or hundreds of kilometers, potentially with significant influence from anthropogenic emissions (Aktypis et al. 2024; Kalkavouras et al. 2021). Currently, observational research on NPF nucleation and growth processes at the atmospheric boundary layer top (BLT) and their contribution to CCN remains limited, which hinders a full understanding of the nucleation mechanisms underlying NPF. Previous studies have observed variations in CCN number concentration ($N_{CCN}$) under different supersaturation (SS) and identified influences from factors such as chemical composition (Wu et al., 2024) and seasonal emission differences (Hirshorn et al., 2022). Over the past three decades, the observational foundation for NPF has been substantially expanded, and numerous models have been developed to describe the process from both mechanistic and empirical perspectives. However, the contribution of NPF to the CCN budget exhibits pronounced spatial heterogeneity. This variability stems largely from the high sensitivity of the subsequent particle growth process—through which newly formed particles evolve into CCN—to local environmental factors, including precursor chemical composition and growth mechanisms (Shen et al., 2016; Zhang et al., 2019). Consequently, despite advances in understanding NPF itself, constraints on the quantitative pathways from nucleation to CCN remain a significant source of uncertainty in aerosol-climate assessments (Kerminen et al., 2012). Current research typically quantifies NPF's enhancement of $N_{CCN}$ by comparing $N_{CCN}$ before and after NPF

events (denoted as $N_{CCN-prior}$ and $N_{CCN-after}$ respectively), using an enhancement factor ($EF_{CCN}$)
generally ranging 0-10 (Liu et al., 2018). Here, $N_{CCN-prior}$ represents the average $N_{CCN}$ during the
two hours preceding an NPF event burst, while $N_{CCN-after}$ denotes the average $N_{CCN}$ from the onset
to the conclusion of NPF's impact on $N_{CCN}$ (Ren et al., 2021; Sun et al., 2021). However, $EF_{CCN}$
primarily quantifies the net enhancement in CCN concentration resulting from an NPF event. While
valuable for assessing the overall impact, this metric does not directly capture the kinetics of the
underlying process, specifically, the rate at which the newly formed particles population grows to
CCN-active sizes. Anthropogenic pollutants in polluted atmospheres directly enhance the
condensational growth rate of newly formed particles by increasing condensable vapor availability,
as demonstrated in urban environments (Dinoi et al., 2023; Kalkavouras et al. 2020; Liu et al., 2018).
These contrasting findings suggest that precursor abundance, atmospheric oxidation capacity, and
background aerosol loading collectively determine whether NPF enhances or suppresses CCN
formation. This underscores the need to focus on regions with complex emission mixtures and
intense human activity, where both natural and anthropogenic drivers strongly interact.
China has emerged as a critical hotspot for studying NPF-to-CCN processes due to its dense urban
clusters and complex interactions between anthropogenic and natural emissions. NPF events occur
frequently in Chinese urban clusters (Chu et al., 2019), including the Yangtze River Delta (YRD).
Yet, their growth to CCN sizes has rarely been examined, and existing studies are largely restricted
to near-surface observations. The YRD area in China, as a globally representative region of intense
anthropogenic emissions, provides abundant species for NPF nucleation and growth processes due
to its high precursor concentrations ($SO_2$, $NH_3$, VOCs, etc.) and active photochemical oxidation
processes (generating gaseous sulfuric acid, gaseous nitric acid, and secondary organic aerosols,
among others) (Qi et al., 2018; Yao et al., 2018). Notably, the BLT in this region serves as a critical
interface connecting polluted air masses with cleaner free tropospheric air, functioning as an
"atmospheric reactor". Under these conditions, the mechanisms through which NPF events
contribute to CCN at the atmospheric BLT may differ significantly from those in surrounding urban
clusters and other high-altitude regions. Studies indicate that high $NH_3$ concentrations (-10 ppbv;
Sun et al., 2023) are frequently observed at this region which promotes an increase in nucleation
rates. Simultaneously, organic acids and nitrates generated from VOC oxidation can form low-
volatility substances, enhancing particle hygroscopic growth capacity (Huang et al.,2024). However,
high precursor concentrations and strong atmospheric oxidation capacity inevitably accompany
higher background aerosol concentrations and more complex chemical compositions. Therefore, it
is critically important to elucidate how atmosphere with strong atmospheric oxidation capacity
under polluted conditions at this BLT environment influence new particle formation and growth
processes, ultimately determining the efficiency of their contribution to CCN production.
This study conducted comprehensive observations at a high-altitude BLT background site in YRD
region in China during spring—a season characterized by frequent NPF events (Qi et al., 2015). By
integrating data on particle number size distributions (PNSD, 2nm-20μm), aerosol chemical
composition, and volatile organic compounds (VOCs) with cluster analysis and model simulations,
we focused on investigating the relationship between NPF and cloud condensation nuclei (CCN).
Specifically, the study aims to quantify the conversion efficiency from NPF to CCN, identify the
mechanisms governing this process under polluted conditions, and propose potential indicators to
improve the representation of CCN sources in regional climate models.
**2. Methodology**
**2.1 Experimental site and Instruments**
A continuous online observation campaign was conducted at the Shanghuang Ecological and
Environmental Observation of the Chinese Academy of Sciences (Shanghuang station; 28.58°N,
119.51°E) from April 19 to May 30, 2024 (Figure S1). The station is located in Jinhua City, Zhejiang
Province, at an elevation of 1128 meters above sea level. It is characterized by mountainous terrain
and forest coverage, representing a typical high-altitude background environment in the YRD region
of China, more details about Shanghuang station could be found in Zhang et al. (2024) and Wang et
al. (2025).
Ambient particles and droplets were initially selected using an advanced aerosol–cloud sampling
inlet system, which alternated between the $PM_1$ cyclone, $PM_{2.5}$ cyclone and total suspended
particulate (TSP) passage every 20 min (Xu et al., 2024). To minimize particle loss within the
sampling system, the relative humidity (RH) at the inlet was maintained below 30% using a Nafion
dryer and a sheath air cycle system. Additionally, diffusion and gravitational losses in the inlet
tubing were corrected based on the tubing shape and flow rate (Baron & Willeke, 2001). The particle
number size distribution (PNSD) from 2.5 nm to 20 μm was continuously measured using an
integrated system. The system consisted of a Neutral Cluster and Air Ion Spectrometer (NAIS, Airel
Ltd.) covering a mobility diameter (dm) range of 2.5-42 nm, a scanning mobility particle sizer
(SMPS, model 3936, TSI Inc.) for 14.5-710 nm (dm) comprising a model TSI3080 electrostatic
classifier and a model TSI 3775 condensation particle counter, and an Aerodynamic Particle Sizer
(APS, model 3221, TSI Inc.) for 0.5–20 μm (aerodynamic diameter, da). Prior to and during the
campaign, regular zero checks and flow-rate verifications were performed using a calibrated
primary flow meter. The NAIS was operated at a sample flow rate of 60 L min⁻¹ to minimize
diffusion losses, with data recorded at 10-min resolution (Mirme and Mirme, 2013). The SMPS was
run with an aerosol-to-sheath flow ratio of 0.3:3.0 L min⁻¹ (1:10), and the APS with an aerosol flow
of 1.0 L min⁻¹ and a sheath flow of 4.0 L min⁻¹ (Liu et al., 2016). Data from the SMPS and APS,
recorded at 5-min resolution, were averaged into hourly spectra and merged into a unified particle
size spectrum matrix (dm: 14.5 nm to 16,000 nm) following the procedure described by Beddows
et al. (2010).
To explore the chemical difference of newly formed particles during their growth processes, the
volatile characteristic of those particles was measured using the Thermal Denuder (TD) -SMPS
system. Volatile analysis helps distinguish between categories of inorganic compounds, such as
nitrates and sulfates and can indirectly provide information on aerosol composition (Schmid et al.,
2002). By comparing the volumes of heated (300°C) and unheated particles, the volatility
characteristics of particles are studied under the assumption that the particles are spherical and
characterized using the volume fraction remaining (VFR) of submicron aerosols. The remaining
semi-volatile components are mainly organic compounds, while components such as sulfates and
nitrates are evaporated at high temperatures, thereby investigating changes in the proportion of
semi-volatile components during the NPF growth process.
CCN number concentration can be measured by CCN counter (Model CCN-100; Deng et al., 2011).
The instrument operated at five supersaturation —0.07%, 0.11%, 0.20%, 0.40%, and 0.80%—with
each level maintained for 15 minutes. To ensure data reliability, measurements were filtered based
on established quality control criteria addressing SS instability within the growth chamber (Rejano
et al., 2021). The total flow rate was maintained at 0.5 L min$^{-1}$, with an aerosol to sheath flow ratio
of 1:10.
The chemical composition of non-refractory submicron particles (NR-PM$_{2.5}$), including organics,
sulfate, nitrate, ammonium, and chloride, was measured using an Aerodyne Time-of-Flight Aerosol
Chemical Speciation Monitor (ToF-ACSM, Li et al., 2023). The instrument sampled ambient air
through the same inlet as the PNSD system, with a flow rate of 0.1 L min$^{-1}$ and a time resolution of
10 minutes. The ToF-ACSM was operated with a capture vaporizer, and its ionization efficiency (IE)
was calibrated at the start of the campaign using 300 nm ammonium nitrate particles. The default
relative ionization efficiencies (RIEs) for nitrate, organics, and chloride (1.1, 1.4, and 1.3,
respectively) were applied (Nault et al., 2023). According to the ion efficiency (IE) calibration
results using ammonium sulfate, the RIE values of ammonium and sulfate were 5.05 and 0.73,
respectively (Zhang et al., 2024). A composition-dependent collection efficiency (CE) was applied
to the raw data to correct for particle losses in the aerodynamic lens, following the parameterization
established by Middlebrook et al. (2012).
The concentrations of major gaseous precursors were measured using the following commercial
analyzers: a pulsed UV fluorescence analyzer (Thermo Scientific, Model 43i) for sulfur dioxide
(SO$_2$), a UV photometric analyzer (Thermo Scientific, Model 49i) for ozone (O$_3$), a
chemiluminescence analyzer (Thermo Scientific, Model 42i) for nitrogen oxides (NO$x$), and a
cavity ring-down spectrometer (Picarro, Model G1103) for ammonia (NH$_3$). Prior to the campaign,
all gaseous analyzers (SO$_2$, O$_3$, NO$x$, and NH$_3$) were calibrated with certified reference gases and
zero air. In addition, routine calibration checks for these gaseous instruments were performed
biweekly throughout the measurement period to ensure continuous accuracy and consistency of the
data of gaseous pollutants. Additionally, PM$_{2.5}$ mass concentrations were measured using a
continuous ambient particulate monitor (Model 5014i, Thermo Scientific, USA), with a PM$_{2.5}$ size
cut-off applied prior to the sampling inlet. Meteorological parameters were recorded during the
measurement period using an automated weather observation system (Milos520, Vaisala, Finland)
positioned adjacent to the PNSD system. A more comprehensive description of the instruments is
available in our previous work (Yang et al., 2021).
**2.2 Data processing of NPF**
Based on their size, atmospheric aerosol particles are commonly grouped into four modes:
nucleation mode (< 20 nm), Aitken mode (20–100 nm), accumulation mode (100–1000 nm), and
coarse mode (> 1 μm). In this study, the number concentration of each mode was obtained by
integrating the measured particle number size distribution over the corresponding diameter interval.
An NPF event is identified when a distinct and sustained (≥ 2 h) burst of nucleation-mode
particles—particularly in the sub-6 nm size range—is observed, followed by a clear growth of the
mode to larger sizes (Dal Maso et al., 2005). Cases failing to meet these criteria were classified as
non-NPF events. The formation rate (J$_{2.5}$), growth rate (GR) and CS were calculated with the

commonly used method (Yang et al., 2021). Recognized as a key contributor to particle nucleation, the concentration of sulfuric acid ($H_2SO_4$) was estimated via a proxy approach proposed by Lu et al. (2019). Additionally, to assess how sulfuric acid ($H_2SO_4$) influences the early-stage particle growth, its contribution to the initial growth rate was quantitatively evaluated using the equation introduced by Nieminen et al. (2010).

**2.3 Calculation of $N_{CCN}$, activation diameter and hygroscopic parameter**

In this study, the $N_{CCN}$ and activation diameter (Da) was calculated by κ-Köhler theory (Petters & Kreidenweis, 2007), which simply link the Da with the supersaturation, is applied as follows, when $κ > 0.1$:

$$\kappa = \frac{4A^3}{27D_a^3 ln^2 S} \qquad (1)$$

$$A = \frac{4\sigma_\omega M_\omega}{RT\rho_\omega} \qquad (2)$$

Here, $\sigma_\omega$ denotes the surface tension of the droplet at the activation point ($\sigma_\omega = 0.072$ J·m⁻²), $M_\omega$ is the molecular weight of water ($M_\omega = 0.018015$ kg·mol⁻¹), T is the temperature of the air parcel, R represents the universal gas constant (R = 8.315 J·K⁻¹·mol⁻¹), and $\rho_\omega$ refers to the density of water ($\rho_\omega = 997.1$ kg·m⁻³). The hygroscopicity parameter κ, which reflects the water affinity of aerosols, is influenced by their chemical composition. In this study, κ was estimated using the Zdanovskii–Stokes–Robinson (ZSR) mixing rule (Stokes & Robinson, 1966), based on chemical volume fractions under the assumption of internally mixed particles, following the approach of Gunthe et al. (2011), as follows:

$$\kappa_{chem} = \sum_i \varepsilon_i \kappa_i \qquad (3)$$

Here, $\kappa_i$ and $\varepsilon_i$ represent the hygroscopicity parameter and volume fraction of each individual dry component in the mixture, respectively. The κ values and corresponding densities (ρ) used in the calculations were adopted from Petters & Kreidenweis (2007) and Topping et al. (2005). While the approach combining the critical dry diameter and bulk aerosol properties may introduce some degree of uncertainty, previous studies have shown that the discrepancy between predicted and measured CCN concentrations remains within an acceptable margin of ±20% under both polluted and pristine atmospheric conditions (Zhang et al., 2017).

**2.4 Quantification the contribution of NPF to CCN**

**2.4.1 Enhancement in $N_{CCN}$**

The enhancement in $N_{CCN}$ attributed to NPF events, referred to as $EF_{CCN}$, was assessed following the methodology described by Kalkavouras et al. (2019) and Ren et al. (2021). The method involves a comparison between the $N_{CCN}$ after and prior to the NPF event:

$$EF_{CCN} = \frac{N_{CCN\text{-}after}}{N_{CCN\text{-}prior}} \qquad (4)$$

Here, $N_{CCN\text{-}prior}$ represents the two-hour average CCN concentration measured before the onset of the NPF event, while $N_{CCN\text{-}after}$ corresponds to the mean value during the period influenced by the

nucleation process. As a simplified approximation, $N_{CCN}$ was estimated by integrating particle number concentration with a particle size larger than the $D_a$. The duration over which NPF contributed to CCN was identified by analyzing changes in the time series of $N_{CCN}$ under each applied supersaturation condition. It is important to note that this approach assumes the background level of CCN remains stable throughout the NPF event, thereby neglecting potential influences from alternative aerosol sources or sinks. As a result, the method provides only an approximate evaluation of the NPF impact on $N_{CCN}$.

**2.4.2 Metric to define the duration of NPF to CCN**

The impact of NPF on CCN has been frequently assessed using metrics such as the CCN enhancement factor ($EF_{CCN}$) as mention in section 2.4.1. More recently, observational studies have conceptualized the timescale of this process by analyzing the interval between the nucleation burst and the subsequent increase in CCN concentration. For instance, Kalkouras et al. (2019) characterized this period through parameters such as $t_{start}$ and $t_{decoupling}$, which effectively capture the climatological time lag of CCN production from NPF events. Still, those methods do not directly deconvolve or quantify the intrinsic, process-level kinetics of the growth path itself. Building upon this foundation, the present study introduces a complementary, process-oriented metric—the "Time Window ($\tau$)"—to further quantify the intrinsic efficiency of CCN production during NPF. While metrics based on observational time lags reflect the net outcome influenced by both growth dynamics and variable background conditions, $\tau$ aims to isolate and quantify the core physical–chemical process: the theoretical time required for a newly formed particle to grow from its initial detectable diameter ($D_0$) to the critical activation diameter ($D_a$) at a given supersaturation. The activation diameter is derived from $\kappa$-Köhler theory, using an effective hygroscopicity parameter ($\kappa$) that represents the chemical composition of the growing nucleation mode. The time window $\tau$ (in hours) is calculated as:

$$\tau = (D_a - D_0)/GR_{nuc} \qquad (5)$$

where $D_a$ is the average critical activation diameter during NPF events (07:00-18:00 LT), $D_0$ is the average diameter of the smallest nucleation mode particles at NPF onset, $GR_{nuc}$ is the average growth rate throughout the NPF growth phase. By directly linking the particle growth rate and its evolving hygroscopicity to the CCN activation threshold, $\tau$ provides a standardized, mechanistic measure that enables comparative analysis of NPF-to-CCN conversion efficiency across diverse atmospheric environments and pollution regimes. This approach more clearly describes the dynamic process in which newly formed particles grow via condensation (increasing dry size and/or altering chemical composition) to the critical size and hygroscopicity required to act as CCN at defined supersaturation, and thus extends current methodologies by offering a more process-explicit framework to evaluate how precursor conditions and chemical pathways modulate the climatic impact of NPF.

**3. Results and Discussion**

**3.1. Characteristic of NPF events**

During the intense campaign, eight NPF events were identified across 39 valid observation days from April 19 to May 30 at the Shanghuang station. Note that bursts in the concentration of freshly nucleated sub-6 nm particles were also observed on April 30 and May 16 (see Figure 1d). However, these two episodes were not classified as NPF events because they occurred at night and were not followed by sustained growth of the nucleation mode to larger sizes, which is a key criterion for defining a full NPF event. Meteorological elements (Figures 1e-f) show that southerly winds dominated during the period of observation, with low average wind speeds (1.9 m·s$^{-1}$). The average relative humidity (RH) during the NPF occurrence time (7:00-18:00 LT) was 63% and 75% for NPF days and non-NPF days, respectively, while the temperature was comparable for NPF and non-NPF days (21.0 and 19.7°C). Thus, there was one NPF event in April and seven in May, resulting in an overall NPF frequency of 21%. This value is higher than the observational values for European high-altitude sites (900 ~ 1200 meters above sea level) during the springtime (Zugspitze Schneefernerhaus: 3%; Hohenpeißenberg: 7%; Sun et al., 2024), while it is similar to nearby urban site (Shanghai: 20%; Xiao et al., 2015) and mountain site in North China (Mountain Tai: 21%; Lv et al., 2018).

The air mass clustering analysis via backward trajectories (Draxler & Hess, 1998) was performed to track the origination of these NPF events, which identified four distinct air mass categories during the observation period (Figure S2). As showed in Figure S1, Cluster 1 represents the polluted air masses affected by North China Plain urban emissions, Clusters 2 and 4 represent the relative clean air mass affected by western and southern urban emissions, Cluster 3 represents the air masses affected by coastal emissions. Combining trajectory analysis with PM$_{2.5}$ mass concentrations, we categorized the eight NPF events into two types: NPF-C events (occurred under clean conditions in Cluster 2-4) and NPF-P events (occurred under polluted conditions in Cluster 1), with average PM$_{2.5}$ of NPF-P events 101% higher that during NPF-C events (12.8 vs 6.4 μg·m$^{-3}$). As showed in Figure 1d, significant variations in 2-6 nm Nucleation mode particles were observed among the eight NPF events, the peak value of which ranged from 246 to 1318 cm$^{-3}$. The average PNSD during NPF-C events and NPF-P events were fitted as the sum of three mode lognormal distributions (Figures 1a-b, Hussein et al., 2005), and revealed that NPF-P events exhibited higher Aitken mode particle concentrations (3978 cm$^{-3}$) than NPF-C events (1980 cm$^{-3}$), while the freshly nucleated sub-6 nm particles were lower in NPF-P (575 cm$^{-3}$ vs. 881 cm$^{-3}$). In addition, the accumulation mode particles were much higher in NPF-P than in NPF-C (881 cm$^{-3}$ vs. 575 cm$^{-3}$). These results indicate that NPF-P event is primarily influenced by regional transportation, whereas NPF-C reflects the background atmospheric conditions at the BLT of the mountain site.

It is worth noting that the NPF event observed on May 5 (NPF-C) occurred during a cloud interstitial period under persistently high relative humidity (> 90%), accompany with a slightly higher formation rate (J$_{2.5}$=0.8 cm$^{-3}$·s$^{-1}$) and growth rate (GR=5.7 nm·h$^{-1}$) compared with the average value of the other NPF-C events (Table 1). We hypothesize that aqueous-phase chemical processes within the preceding cloud were pivotal. A mechanism analogous to the "post-fog growth" reported in the Arctic may be at play, whereby in-cloud reactions generate semi-volatile organic compounds (SVOCs) that later condense onto particles (Kecorius et al., 2023). While direct measurements of the specific SVOCs are not available, the elevated concentration of isoprene-a key biogenic precursor-on that day (0.3 ppbv compared to the 0.2 ppbv average for other NPF-C events) provides

indirect support for enhanced biogenic activity and potential secondary organic aerosol formation pathways. Following cloud dissipation, these cloud-generated condensable vapors were released and, under sustained high humidity, rapidly condensed onto the newly formed nucleation-mode particles. This organic-dominated condensation likely surpassed the nitrate-driven growth observed in other events, facilitating sustained particle growth and enabling a larger fraction of the population to surpass the activation diameter and reach CCN sizes.

**3.2 Diurnal Comparison of Key Drivers and NPF Metrics between Clean and Polluted Events**

To elucidate the factors driving distinct NPF behaviors, this section presents a diurnal comparison of key parameters between clean (NPF-C) and polluted (NPF-P) event days. As shown in Figure 2a, the average formation rate ($J_{2.5}$) during NPF-P events was 2.4 $cm^{-3}\,s^{-1}$, approximately 3.6 times higher than during NPF-C events (0.7 $cm^{-3}\,s^{-1}$). The peak $J_{2.5}$ in NPF-P events (6.2 $cm^{-3}\,s^{-1}$ at 12:00 LT) was also higher and occurred one hour later than the peak in NPF-C events (1.8 $cm^{-3}\,s^{-1}$ at 11:00 LT). The most pronounced enhancement—a fivefold increase—was observed at 10:00 LT (2.5 vs. 0.5 $cm^{-3}\,s^{-1}$). While the average gaseous sulfuric acid ($H2SO4$) concentration was 23 % higher in NPF-P events ($8.1 \times 10^{6}\,cm^{-3}$) and the condensation sink (CS) was also elevated (0.013 vs. 0.008 $s^{-1}$ for NPF-C), the significantly stronger formation and growth rates indicate that enhanced production of condensable vapors from anthropogenic pollution was sufficient to overcome the increased sink strength, enabling intense NPF—a phenomenon documented in other polluted environments (Yang et al., 2021). Crucially, the 23 % difference in $[H_2SO_4]$ alone cannot account for the ~3.6-fold difference in $J_{2.5}$. Ammonia ($NH_3$) played a critical role in this enhanced nucleation. The average $NH_3$ concentration during NPF-P events (8.1 ppbv) was approximately twice that during NPF-C events (4.1 ppbv; Figure 2c). This elevated $NH_3$ level, coinciding with higher $H_2SO_4$, likely contributed to the enhanced nucleation rates observed under polluted conditions by stabilizing sulfuric acid clusters.

Concurrently, NPF-P events exhibited a higher event-average of background ozone ($O_3$) concentration (27.7 ppbv vs. 19.9 ppbv for NPF-C). Although the $O_3$ difference narrowed during the peak nucleation period (10:00–12:00 LT)—suggesting its primary role is in maintaining an enhanced oxidative environment conducive to precursor oxidation rather than directly driving the instantaneous nucleation burst—the difference expanded again after 15:00 LT, reaching a maximum in the late afternoon (18:00 LT; Figure 2d). This later period coincides with the sustained particle growth phase, where a stronger oxidative capacity likely facilitates the production of low-volatility condensable vapors, thereby influencing condensational growth. Correspondingly, the average particle growth rate (GR) during NPF-P events was 6.8 $nm\,h^{-1}$, which is 24% higher than during NPF-C events (5.5 $nm\,h^{-1}$; Figure 2g). The overall elevated GR is consistent with a greater abundance of condensable vapors (e.g., nitrate and photochemically generated organics), which are discussed in the following sections. Compared to typical values reported for a remote boreal forest site (Hyytiälä, Finland: $J_3$= 0.4 $cm^{-3}\,s^{-1}$, GR = 2.3 $nm\,h^{-1}$; Kerminen et al., 2018), the formation and growth rates observed at our site are higher by 275% and 126%, respectively. Our values are close to those reported for other Chinese high-altitude background sites like Mount Tai ($J_3$= 1-2 $cm^{-3}\,s^{-1}$;

Shen et al., 2019), Mount Heng ($J_{15}$ = 0.15-0.45 $cm^{-3}$ $s^{-1}$; Nie et al., 2014), and Mount Yulong ($J_3$ = 1.33 $cm^{-3}$ $s^{-1}$; Shang et al., 2018). These differences suggest that the intensity of an NPF event can vary significantly depending on the atmospheric conditions and the regional transport processes involved.

To investigate the chemical differences driving nanoparticle growth during the two types of NPF events, the diurnal variations of chemical components (organics, sulfates, nitrates, ammonium, chlorides, and black carbon) were analyzed during NPF evolution (Figures 2h-i). The results show that during NPF-P events, mass concentrations of all major chemical components increased alongside particle growth, with organics and nitrates exhibiting the most pronounced and sustained enhancement (Figures 2h-i). In contrast, NPF-C events displayed weaker and less persistent increases. While organics dominated the non-refractory $PM_{2.5}$ (NR-$PM_{2.5}$) mass fraction (accounting for more than half) during the growth phase in both event types, the chemical evolution pathways diverged significantly under anthropogenic influence. The stronger nitrate growth in NPF-P events can be attributed to a more favorable chemical environment. These events were characterized by significantly higher concentrations of $NO_2$ and $NH_3$ (Figure 2c). Photochemical modeling indicates that elevated $NO_2$ under stronger solar radiation leads to enhanced production of gaseous nitric acid ($HNO_3$) (Figure S3). In the presence of abundant $NH_3$, this $HNO_3$ efficiently partitions to the particle phase via neutralization, forming ammonium nitrate (Wang et al., 2022). This process explains the more than fivefold increase in nitrate peak concentrations during the later growth stages of NPF-P events, where nitrate became a key driver for sustained condensational growth. Similarly, the more substantial organic mass increase during NPF-P events is linked to enhanced secondary organic aerosol (SOA) formation (Shi et al., 2016). Higher daytime $O_3$ concentrations (Figure 2d) suggest a more intense oxidative environment, which promotes the photochemical oxidation of volatile organic compounds (VOCs). Coupled with elevated ambient VOC levels (e.g., isoprene), this leads to the production of more low-volatility oxygenated organic molecules that readily condense onto growing particles (Kulmala et al., 2012). Therefore, the synergistic enhancement of nitrate and organic precursors under polluted, transport-influenced conditions provides a robust chemical explanation for the faster and more sustained particle growth observed during NPF-P events compared to NPF-C events.

Previous field studies have highlighted the importance of organics for new particle growth in remote regions (Pierce et al., 2012). Recent comprehensive analyses from multiple European cities further support this view, demonstrating that the growth of nucleated particles is often driven by the condensation of semi-volatile organic compounds (Trechera et al., 2023). Our findings indicate that in anthropogenically influenced mountain regions, nitrate— primarily as ammonium nitrate ($NH_4NO_3$)—can serve as a competitive source of low-volatility condensable vapor, partially substituting for organics in driving the mass growth of new particles. This occurs under conditions of elevated $NO_2$ and $NH_3$, where efficient photochemical production and gas-to-particle partitioning of $NH_4NO_3$ are favored. While the strong hygroscopicity of nitrate plays a secondary role by increasing the particle's wet size (and thus potentially enhancing condensation efficiency under high relative humidity), its primary contribution to growth is through direct vapor condensation.

It should be noted that the analysis of chemical drivers for particle growth in this study relies on the bulk non-refractory $PM_{2.5}$ (NR-$PM_{2.5}$) composition measured by the ToF-ACSM. While CCN

activation at the studied supersaturations primarily involves particles in the Aitken and smaller
accumulation modes (< 200 nm), we assert that the bulk $PM_{2.5}$ composition serves as a valid proxy
for the condensing vapors during sustained NPF events under our background conditions. This is
supported by the fact that during such events, the growth of the nucleation mode is the dominant
source of new aerosol mass in the submicron range. Previous study indicates that changes in bulk
organic and inorganic mass concentrations correlate well with the condensational needs of growing
nanoparticles, making bulk composition a practical and informative metric for identifying dominant
growth pathways (Vakkari et al., 2015). We acknowledge that size-dependent compositional
differences may exist and represent an important avenue for future research with size-resolved
instrumentation.
**3.3 Potential formation mechanism of NPF-C and NPF-P events**
Gaseous sulfuric acid is recognized as an important specie in nucleation across NPF events (Gracia
et al., 2024). The correlation coefficients (R) between $J_{2.5}$ and $[H_2SO_4]$ were 0.77 for NPF-C events
and 0.87 for NPF-P events (Figure 3b). This positive dependence of the nucleation rate on sulfuric
acid concentration is consistent with observations from remote background sites, though the strength
of the correlation varies with the degree of anthropogenic influence (Kulmala et al., 2013). At
pristine sites such as Hyytiälä, the correlation is often moderated by the co-involvement of biogenic
organic vapors and ions (Kulmala et al., 2025), whereas at background sites in China affected by
regional pollution transport, stronger correlations between nucleation and $[H_2SO_4]$ was typically
observed (Gao et al., 2025). The high correlations observed here (R = 0.77–0.87) align with the latter
pattern, reinforcing that our mountain-top station, although a background site, experiences
substantial anthropogenic influence that shapes the nucleation mechanism. However, the moderate
difference in $[H_2SO_4]$ alone cannot explain the large difference in $J_{2.5}$ between event types
(Section 3.2). Previous studies have also indicated that binary $H_2SO_4$–$H_2O$ nucleation cannot fully
account for atmospheric NPF rates (Kirkby et al., 2011). This points to the importance of additional
compounds that stabilize $H_2SO_4$ clusters and modulate nucleation efficiency. Previous field and
chamber studies also proposed that the gaseous species such as ammonia (Kulmala et al., 2013;
Kürten et al., 2019) and amines (Metzger et al., 2010; Yao et al., 2018) also promote the nucleation.
The elevated $NH_3$ concentrations measured during NPF-P events (Figure 2b) thus provide a
plausible explanation for their higher nucleation rates despite a less-than-proportional increase in
$[H_2SO_4]$.
To explore the nucleation mechanism in the atmospheric boundary layer top, the relationship
between $J_{2.5}$ and $[H_2SO_4]$ was analyzed for NPF-P and NPF-C events and compared with results
from CLOUD chamber experiments, which delineate pathways for $H_2SO_4$–$NH_3$–$H_2O$ and $H_2SO_4$–
dimethylamine (DMA)–$H_2O$ nucleation (Kürten et al., 2019; Almeida et al., 2013). As shown in
Figure 3a, our measured formation rates (solid circles: NPF-P; hollow circles: NPF-C) fall within
the $[H_2SO_4]$ range spanned by these two mechanisms in the chamber. Achieving the observed $J_{2.5}$
would require either higher DMA levels or higher $NH_3$ concentrations than those set in the specific
CLOUD runs. Given the lack of significant DMA sources in the region (e.g., textile or industrial
activities; Chang et al., 2022), ambient $NH_3$ (average ~5 ppbv during NPF) is the more plausible
stabilizing base. However, the CLOUD experiments have not yet performed under similar
atmospheric conditions as our field observation (e.g. higher $NH_3$ levels exceed 1ppbv) (Kürten et
al., 2019). Thus, to evaluate the formation mechanism under rich-$NH_3$ conditions representative of
our site, we performed simulations using the MALTE-BOX model (Boy et al., 2006; McGrath et
al., 2012), which couples the Atmospheric Cluster Dynamics Code (ACDC). Input parameters were
set to the average conditions during NPF events: condensation sink (CS) = 0.010 s⁻¹, [NH_3] = 5 ppbv,
RH = 66%, T = 293 K, and pressure = 883 hPa. The model calculates the formation rate for clusters
growing past a critical size as a function of [$H_2SO_4$]. The simulation results are shown as the yellow
line and gray uncertainty band in Figure 3a. Most of our measured $J_{2.5}$ data points fall within or near
the model-predicted band, indicating that $H_2SO_4$-$NH_3$ nucleation is a quantitatively plausible
mechanism under the observed conditions. The model predictions tend to be slightly higher than the
measured rates. This discrepancy may arise because the model's initial cluster definition (e.g., a
$(H_2SO_4)_5(NH_3)_5$ cluster corresponding to ~1.07 nm; Huang et al., 2016) effectively simulates
formation at a smaller size than our observational threshold ($J_{2.5}$), and potential uncertainties in
cluster binding energies or the omission of other stabilizing species (e.g., organic vapors) in the
simulation. Nevertheless, the general agreement supports the conclusion that ammonia-enhanced
sulfuric acid nucleation is a dominant pathway at this site.
Independent support for the role of ammonia comes from the field-observed correlations. A
pronounced linear relationship exists between $J_{2.5}$ and the product of $H_2SO_4$ and $NH_3$ concentrations
(Figure 3c). The Pearson correlation coefficient (R) for $J_{2.5}$ versus [$H_2SO_4$]×[$NH_3$] ranges from 0.79
to 0.92, notably higher than the correlation of $J_{2.5}$ with [$H_2SO_4$] alone (R = 0.77-0.87, Figure3b).
This enhanced correlation when $NH_3$ is included as a co-variable has been observed in other polluted
environments; for example, wintertime measurements in Shanghai reported a tighter relationship
between $J_{1.34}$ and [$NH_3$] ($R^2$=0.62) than with [$H_2SO_4$] ($R^2$= 0.38) (Xiao et al., 2015). Together, the
consistency between our observations and the MALTE-BOX simulations, combined with the strong
field-based correlation that explicitly includes $NH_3$, provides robust evidence that ammonia plays a
key role in enhancing sulfuric acid-driven nucleation at this mountain-top site.

### 3.4. Oxidation-driven acceleration of NPF-to-CCN

This section aims to elucidate the relationship between the growth processes of the two types of
NPF events and their efficiency in forming CCN. To quantify the CCN production from NPF events,
the $N_{CCN}$ was calculated. Since supersaturation (SS) cannot be measured directly at the site, we
employed a sensitivity approach using two representative SS values. These values were selected
based on prior aircraft measurements in the regional background atmosphere, which reported a
range of 0.1-0.5% (Gong et al., 2023). We performed calculations for SS=0.2% and SS=0.4%,
encompassing a common in-cloud condition and a higher activation threshold. For each SS, the
critical activation diameter ($D_a$) was derived using κ-Köhler theory, with the hygroscopicity
parameter ($\kappa$) estimated from the measured particle chemical composition (Bougiatioti et al., 2011),
adjusting for local altitude. The calculated $N_{CCN}$ for both SS levels was then compared with observed
$N_{CCN}$ to evaluate the parameterization's performance and to analyze the SS-dependence of CCN
production efficiency.
**3.4.1 Chemical Drivers of Varied Hygroscopicity and Critical Diameter**
The critical diameter for CCN activation ($D_a$) exhibited a strong dependence on supersaturation (SS),
as theoretically expected. For the studied NPF events, $D_a$ at SS=0.4% was substantially lower than
at SS=0.2%. Under the lower SS condition (0.2%), $D_a$ varied from 111 to 129 nm, with a higher
average in polluted (NPF-P) events (126 nm) compared to clean (NPF-C) events (120 nm). This
difference correlated with a lower average hygroscopicity parameter ($\kappa$) for NPF-P events (0.18)
than for NPF-C events (0.21), originating from a higher organic mass fraction (77% vs. 65 %). At
the higher SS of 0.4%, the average $D_a$ decreased to approximately 80 nm (NPF-P) and 76 nm (NPF-
C), yet the inverse relationship between $D_a$ and $\kappa$ persisted.
The dependence of $D_a$ on supersaturation has significant implications for NPF-driven CCN
production. At a higher SS of 0.4%, the substantially reduced critical diameter shortens the required
growth trajectory and timescale, allowing particles to become CCN-active more rapidly in
environments with elevated supersaturation. Consequently, the net CCN enhancement ($EF_{CCN}$)
during NPF was systematically greater at SS=0.4% than at 0.2%. Notably, while pollution-enhanced
CCN production was evident at both SS levels, the relative enhancement of NPF-P over NPF-C
events was more pronounced at the lower SS (0.2%). This indicates that the chemically processed,
faster-growing particles in polluted air masses are particularly effective at overcoming the greater
activation barrier (larger $D_a$) at low SS. In addition, the chemical composition itself was shaped by
the precursor environment. Although the condensation sink (CS) was elevated during NPF-P events
(0.013 $s^{-1}$ vs. 0.008 $s^{-1}$ for NPF-C), which typically suppresses nucleation, significantly higher
concentrations of gaseous sulfuric acid ($H_2SO_4$) and nitric acid ($HNO_3$) were present (Figure 2b,
S3). This indicates that the enhanced production of condensable inorganic vapors under pollution
transport was sufficient to overcome the increased vapor sink, thereby promoting intense nucleation
and growth. Notably, $HNO_3$ played a dual role. First, it contributed directly to particle growth via
the formation of ammonium nitrate. Second, as a strong oxidant, $HNO_3$ (often in conjunction with
other oxidants like OH) enhances the atmospheric oxidation of volatile organic compounds (VOCs),
promoting the formation of low-volatility oxygenated organic compounds (LV-OOCs). The
condensation of these LV-OOCs further increases the organic mass fraction of the growing particles.
This pathway, where $HNO_3$ indirectly promotes the condensation of low-$\kappa$ organic material,
provides a chemical mechanism for the observed suppression of average particle hygroscopicity ($\kappa$)
in NPF-P events. Support for this mechanism comes from Thermal Denuder (TD) measurements,
which showed a higher volume fraction remaining (VFR) at 300 °C for NPF-P events (Figure 4d),
indicating a greater proportion of low-volatility/non-volatile (refractory) material consistent with a
processed, low-$\kappa$ organic fraction.

**3.4.2 Temporal Evolution of Particle Growth and CCN Activation Efficiency**

The efficiency with which newly formed particles evolve into CCN is governed by the interplay between their dynamic growth and concurrent changes in hygroscopicity, as illustrated in Figure 4. During the initial hours of NPF events, particle volatility analysis reveals an elevated non-volatile fraction (high VFR; Figure 4d). This indicates a substantial presence of low-hygroscopicity material, such as highly oxidized organics, which lowers the effective particle hygroscopicity ($\kappa$). As a direct consequence, the critical activation diameter ($D_a$) peaks at $\sim$124 nm for NPF-C and $\sim$129 nm for NPF-P events in this phase (Figs. 4a, S4a), since less-hygroscopic particles require a larger dry size to activate.

The diurnal evolution of the particle population further elucidates the transition from nucleation to CCN production. Total particle number concentration ($N_{CN}$) begins a rapid increase after $\sim$07:00 LT, driven by the nucleation burst (Figure 4b). Although $N_{CCN}$ starts to rise concurrently, the explosive production of small nucleation-mode particles initially causes the activation ratio (AR = $N_{CCN}/N_{CN}$) to decline, reflecting the time required for growth to CCN-active sizes. $N_{CCN}$ subsequently peaks around 09:00–10:00 LT, approximately 2–3 hours after the $N_{CN}$ surge, marking the period when a substantial fraction of newly formed particles has grown sufficiently. After $\sim$14:00 LT, as growth processes intensify (indicated by high GR), an increasing number of particles reach $D_a$, and the AR begins a gradual recovery (Figures S4b-c).

Underlying these dynamics are distinct chemical drivers that shape both growth and volatility. Organic components dominated the particle composition, accounting for over 60 % of NR-PM$_{2.5}$ mass on average across events (Figure 4c). The VFR in the 14-80 nm size range was 10-20 % (Figure 4d), significantly higher than values reported for polluted urban Beijing ($\sim$5 %; Wu et al., 2017). Because heating to 300 °C effectively removes volatile inorganic salts and semi-volatile organic compounds, a higher VFR primarily reflects a greater abundance of low-volatility organic compounds (LVOCs). At our background site, where local combustion influence is minimal, this points to a more aged, oxidized organic aerosol component (Ehn et al., 2014; Jimenez et al., 2009), consistent with the observed lower $\kappa$ and higher $D_a$.

A size-resolved perspective reveals how growth pathways shift as particles mature. The contribution of gaseous sulfuric acid to the growth rate (GR) declines sharply with increasing particle size (Figure 4d). In NPF-C events, for instance, its contribution drops from $\sim$20 % in the 2-6 nm bin to < 8 % in the 15-20 nm bin, indicating that other condensable vapors become dominant for subsequent growth (Yang et al., 2021; Zhu et al., 2023). In NPF-P events, the sulfuric acid contribution is consistently lower than in NPF-C events (e.g., 6 % vs. 8 % in the 9–12 nm bin), suggesting a greater role for alternative vapors—such as nitrate and oxidized organics—under polluted conditions. Concurrently, the non-volatile fraction (1-VFR) increases with particle diameter, approaching 90 % in the 60–120 nm bin. This trend underscores the growing importance of low-volatility material in driving particles to CCN sizes as they mature. The slightly lower non-volatile VFR during NPF-P events further highlights the significant contribution of organic components at the boundary-layer top, which helps explain the persistently high $D_a$ observed during the initial stage of NPF.

### 3.4.3 Quantitative Assessment of NPF-to-CCN Conversion Efficiency and Kinetics

The distinct chemical pathways observed under clean and polluted conditions translate into

quantifiable differences in the efficiency and speed of CCN production. To evaluate the net impact of NPF on the CCN budget, we first employed the established CCN enhancement factor ($EF_{CCN}$), which characterizes the relative increase in CCN number concentration following an NPF event. The $EF_{CCN}$ revealed a strong dependence on pollution level: the average $EF_{CCN}$ for polluted (NPF-P) events (1.8) was 161% higher than for clean (NPF-C) events (0.7), confirming that precursor enrichment has a significant promoting effect on CCN formation. This trend aligns with other studies where higher $EF_{CCN}$ typically correlates with enhanced anthropogenic influence (Rejano et al., 2021). However, it is important to note that $EF_{CCN}$ represents an aggregate outcome integrating contributions from both newly formed and pre-existing particles, where the latter can introduce substantial deviations in estimated CCN enhancements (Kalkavouras et al., 2019).

While $EF_{CCN}$ quantifies the net CCN enhancement, it does not capture the dynamics of the conversion process. To address this, we introduced the "Time Window ($\tau$)", which quantifies the duration required for newly formed particles to grow from their initial size to the critical activation diameter ($D_a$). This kinetic metric showed substantial variability across events (15.1–22.2 h), with the average $\tau$ for NPF-P events (16.4 h) being 17% shorter than for NPF-C events (19.8 h). This directly demonstrates that polluted conditions accelerate the NPF-to-CCN conversion. A lower $\tau$ value, driven by strong atmospheric oxidation capacity, accelerates CCN conversion within a shorter timeframe—coupling effectively with daytime boundary layer cloud cycles to boost CCN supply efficiency (Kommula et al., 2024). The relationship between these metrics is illustrated in Figure 5a, where a shorter $\tau$ (faster growth) correlates strongly with a higher $EF_{CCN}$ (greater CCN enhancement). This inverse relationship underscores that the efficiency of CCN production is intrinsically linked to the speed of particle growth. Further analysis linked these metrics to pollution intensity. Figure 5b shows a positive correlation between $EF_{CCN}$ and local $PM_{2.5}$ mass concentration, indicating that elevated precursor concentrations enhance NPF growth and intensify nanoparticle conversion to CCN. Figure 5c reveals a clear negative correlation between $\tau$ and the particle growth rate (GR), confirming that faster growth universally shortens the conversion timescale, regardless of activation state and initial diameter.

The accelerated kinetics in NPF-P events can be attributed to the synergistic effects of elevated precursor concentrations and enhanced atmospheric oxidation. While transported oxidation products like highly oxygenated organic molecules (HOMs) may slightly suppress particle hygroscopicity, the concurrent surge in condensable inorganic vapors—particularly ammonium nitrate, as evidenced by the growing nitrate fraction in the afternoon and evening (Figs. 4e-f)—provides a powerful and sustained driver for rapid condensational growth. Once partitioned into the particle phase, ammonium nitrate increases the overall particle hygroscopicity($\kappa$). This physicochemical effect counteracts the hygroscopicity suppression by organics, effectively lowering the critical activation diameter ($D_a$) at a given supersaturation and facilitating the activation of growing particles into CCN. This combination of factors enables particles to overcome the initial hygroscopicity limitation and efficiently reach CCN sizes. In contrast, under cleaner conditions (NPF-C), the nitrate fraction remains low and stable (Figure 4e), signifying a minimal role in the growth process and leading to slower growth that extends the CCN conversion window.

To assess the general applicability of the $\tau$–GR relationship, we conducted parallel analyses on

published datasets from multiple European sites representing diverse environmental regimes under comparable supersaturation conditions (SS = 0.4%). The sites included: Leipzig-TROPOS (LTR, urban background), Bösel (BOS, urban background), Melpitz (MEL, regional background), Neuglobsow (NEU, regional background), Hohenpeißenberg (HPB, high-altitude), Schauinsland (SCH, high-altitude), and Zugspitze Schneefernerhaus (ZSF, high-altitude; Sun et al., 2024). A statistically significant negative correlation between aerosol lifetime ($\tau$) and growth rate (GR) was consistently observed across all sites, indicating that enhanced growth kinetics promote accelerated cloud condensation nuclei (CCN) activation. This consistent pattern confirms that the inverse $\tau$–GR relationship is a robust feature across varied atmospheric environments, extending beyond the specific conditions of our primary study site.

In conclusion, while some studies suggest CCN production from NPF can be suppressed in intensely polluted urban cores, our findings demonstrate that in background regions receiving aged pollution plumes, the transported pollutants create a chemical environment that simultaneously enhances nucleation rates and accelerates subsequent particle growth. This dual effect is quantified by a significantly higher $EF_{CCN}$ and a substantially shortened Time Window ($\tau$), leading to a more efficient and faster coupling between NPF and CCN production. This oxidation-driven acceleration mechanism represents a key pathway through which anthropogenic emissions can intensify aerosol-cloud interactions in downwind regions.

**4. Conclusion**

Our intensive mountain-top observations in the YRD demonstrate that polluted air masses substantially accelerate NPF and its conversion to cloud condensation nuclei (CCN). Across eight identified NPF events, those under polluted conditions (NPF-P) exhibited a 360% higher nucleation rate ($J_{2.5}$ = 2.5 vs. 0.7 cm$^{-3}$ s$^{-1}$) and a 24% faster growth rate (GR = 6.8 vs. 5.5 nm h$^{-1}$) compared with clean events (NPF-C). These enhancements were accompanied by elevated $NH_3$ concentrations (8.1 vs. 4.1 ppb) and higher gaseous $H_2SO_4$ (8.2×10$^6$ cm$^{-3}$, 23% higher than NPF-C), confirming ternary $H_2SO_4$–$NH_3$–$H_2O$ nucleation as the dominant mechanism, consistent with MALTE-BOX simulations. The polluted events further yielded a markedly larger CCN enhancement factor ($EF_{CCN}$ = 1.6 vs. 0.7 in clean cases), reflecting the strong contribution of anthropogenic oxidation products and secondary nitrate condensation. Using the novel "Time Window ($\tau$)", we show that polluted air masses shortened the NPF–to–CCN conversion timescale by 17% ($\tau$ = 16.4 h vs. 19.8 h), enabling nascent particles to reach activation sizes within the diurnal cloud cycle. Notably, nitrate accumulation during afternoon growth phases sustained high GR, compressing $\tau$ and ensuring efficient CCN supply. These results together suggest that cross-regional pollutant transport enhances precursor abundance, boosts atmospheric oxidation capacity, and accelerates both the magnitude and timing of CCN production at the BLT. Collectively, these results suggest that cross-regional pollutant transport enriches precursor concentrations, elevates the atmospheric oxidation capacity, and thereby enhances both the magnitude and advances the timing of CCN production at the boundary layer top. Crucially, while previous studies have indicated that intense local pollution can suppress CCN formation from NPF, our findings demonstrate that in oxidizing, transport-influenced environments such as the one studied here, aged pollution plumes can instead amplify CCN yields.

Accurately representing these oxidation-driven growth pathways in atmospheric models is therefore essential for constraining aerosol-cloud-climate feedbacks in rapidly developing regions.

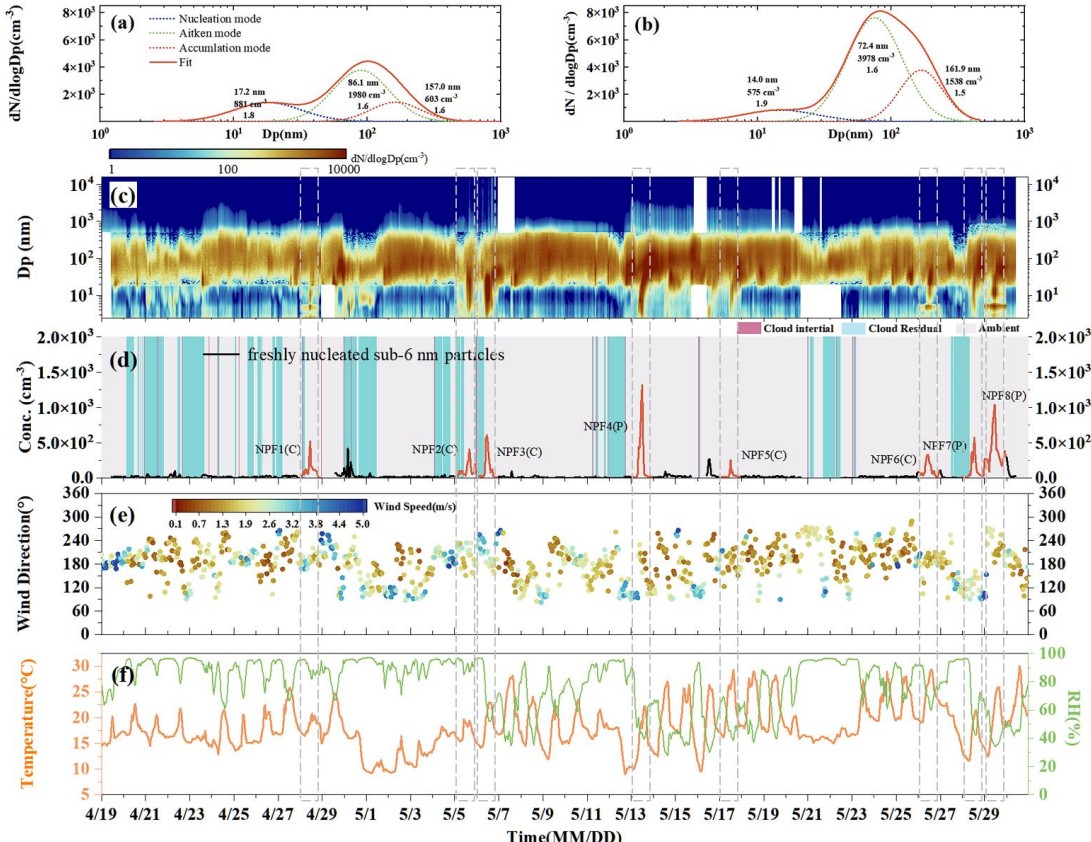

Figure 1: Overview of atmospheric conditions and new particle formation (NPF) events at the mountain-top station. The dashed-line frame represents the NPF events days. (a-b) Lognormal-fitted particle number size distributions for representative (a) clean (NPF-C) and (b) polluted (NPF-P) NPF events. Fitted modes are color-coded: nucleation (<20nm, blue), Aitken (20-100nm, green), and accumulation (100-1000nm, orange). (c) Time series of observed particle number size distributions (dN/dlogDp) during the entire campaign. (d) Temporal evolution of particle types: cloud interstitial (dark red), cloud residual (light blue), and non-cloud periods (Ambient, light gray). The occurrence of sub-6nm particles (fresh nucleation) is overlaid as red lines, highlighting identified NPF event days. (e) Wind direction time series, where color intensity represents wind speed magnitude. (f) Time series of temperature and relative humidity.

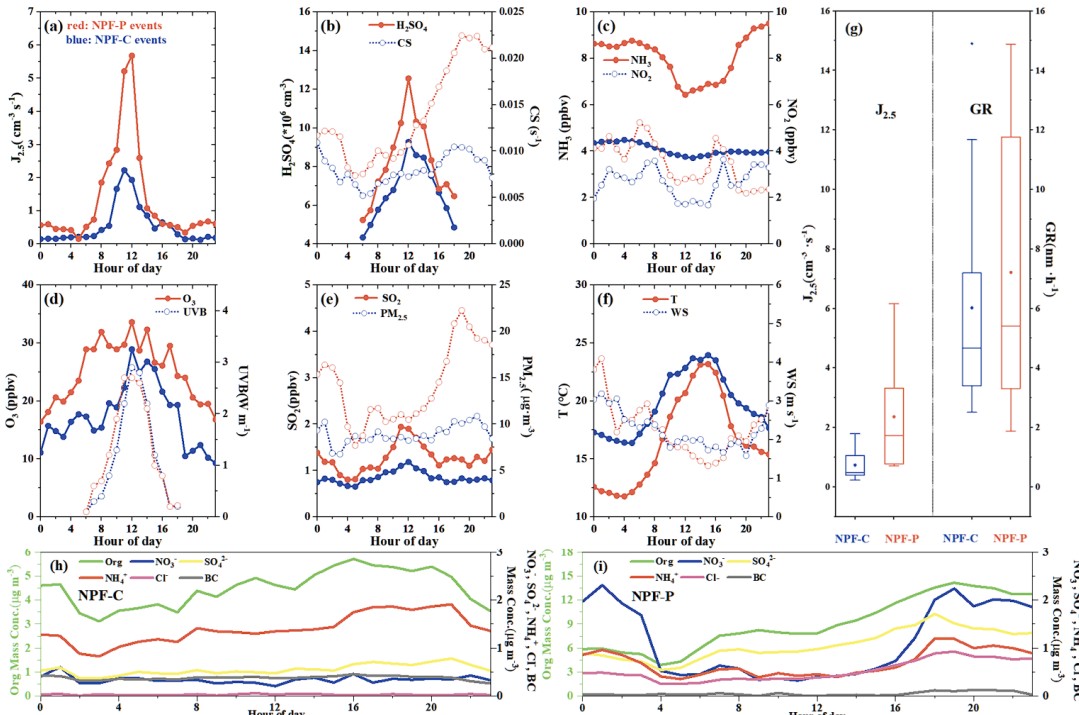


Figure 2: Diurnal comparison of key parameters and new particle formation (NPF) metrics between clean
(NPF-C) and polluted (NPF-P) event days. (a) formation rate ($J_{2.5}$); (b) $H_2SO_4$ concentration and
condensation sink (CS); (c) $NH_3$ and $NO_2$ concentration (d) $O_3$ concentrations and UV-B radiation
intensity; (e) $SO_2$ concentration and $PM_{2.5}$ mass concentration; (f) Temperature (T) and wind speed (WS);
(g) Box plots of formation rate ($J_{2.5}$) and growth rate (GR), where boxes show the interquartile range
(25th-75th percentile), internal lines denote the median, dots represent the arithmetic mean, and whiskers
extend to the 10th and 90th percentiles. (h-i) Mean diurnal profiles of non-refractory $PM_{2.5}$ chemical
composition (organics, sulfate, nitrate, ammonium, chloride) and black carbon (BC) mass concentration
for (h) NPF-C and (i) NPF-P events.

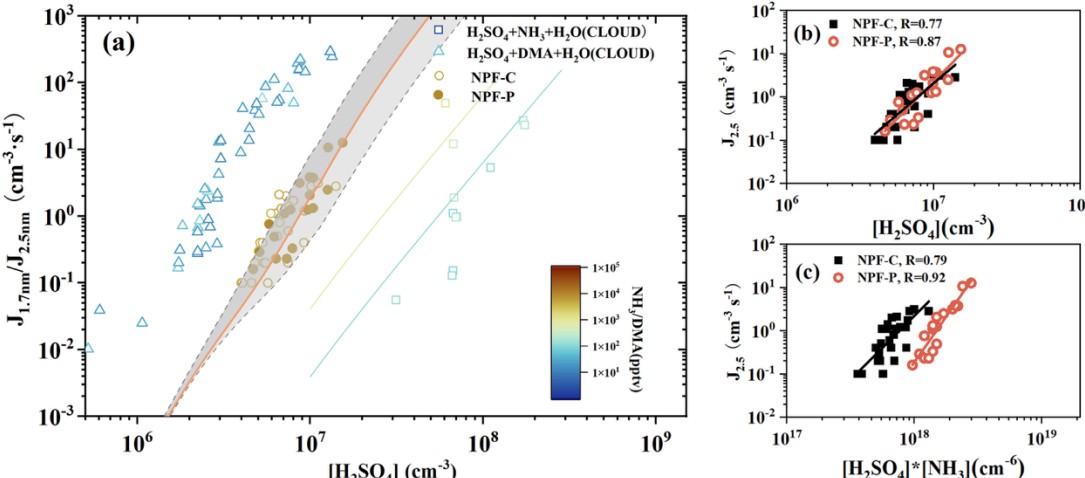


Figure 3: Nucleation mechanism analysis at Shanghuang station. (a) Comparison of formation rates as a
function of $H_2SO_4$ concentration among field observations, CLOUD chamber experiments, and
theoretical predictions. Field measurements are presented as the 2.5 nm formation rate ($J_{2.5}$; colored

circles: hollow for NPF-C events, solid for NPF-P events). These are compared with the 1.7 nm formation rate ($J_{1.7}$; squares and triangles) from CLOUD experiments conducted at 278 K and 38% RH under controlled precursor conditions: $H_2SO_4$-$NH_3$-$H_2O$ ternary nucleation (squares, $NH_3$=0.1 ppbv and 1 ppbv) and $H_2SO_4$-DMA-$H_2O$ ion-mediated nucleation (triangles, DMA=13-140 pptv) (Kürten et al., 2019; Almeida et al., 2013). DMA denotes dimethylamine. Color gradients indicate $NH_3$ (blue) and DMA (red) mixing ratios in the chamber. The yellow line shows the MALTE-BOX model prediction for $H_2SO_4$ nucleation with 5 pptv $NH_3$; the gray band represents the uncertainty in cluster binding energy ($\pm 1$ kcal mol$^{-1}$). (b) Formation rates ($J_{2.5}$) versus $H_2SO_4$ concentration for NPF-C (black squares) and NPF-P (red hollow circles) events. (c) Formation rates ($J_{2.5}$) as a function of the $H2SO4$ and $NH3$ concentration for NPF-C (black squares) and NPF-P (red hollow circles), with Pearson correlation coefficients (R) indicated.

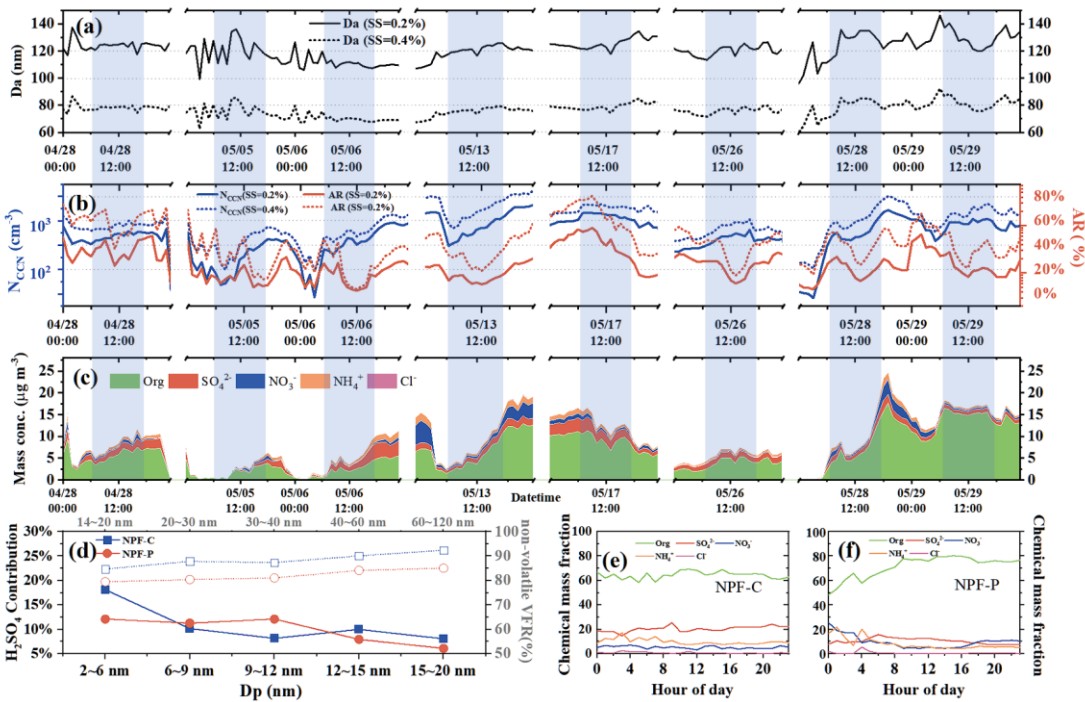

Figure 4: CCN-related parameters and chemical compositions across eight NPF events. (a) The solid line and the dashed line denote to the activation diameters at supersaturation (SS=0.2%) and supersaturation (SS=0.4%) during eight NPF events, respectively. (b) Temporal evolution of $N_{CCN}$ (blue solid line and blue dashed line) and its activation ratio (AR = $N_{CCN}/N_{CN}$, red solid and solid line). The solid line represents SS=0.2% and the dashed line represents SS=0.4%. (c) Time-resolved mass concentrations of particulate chemical constituents (organics, sulfate, nitrate, ammonium and chlorine) during the eight NPF events. (d) Solid line represents the fractional contribution of $H_2SO_4$ to GR within 2 ~ 20 nm particles; dashed line represents the non-volatile volume fraction remaining (1-VFR) in the 14 ~ 120 nm size bin. The blue line denotes to NPF-C events and blue line denotes to NPF-P events. (e-f) Diurnal variations in mass fraction contributions of chemical constituents during NPF-C and NPF-P events, respectively.

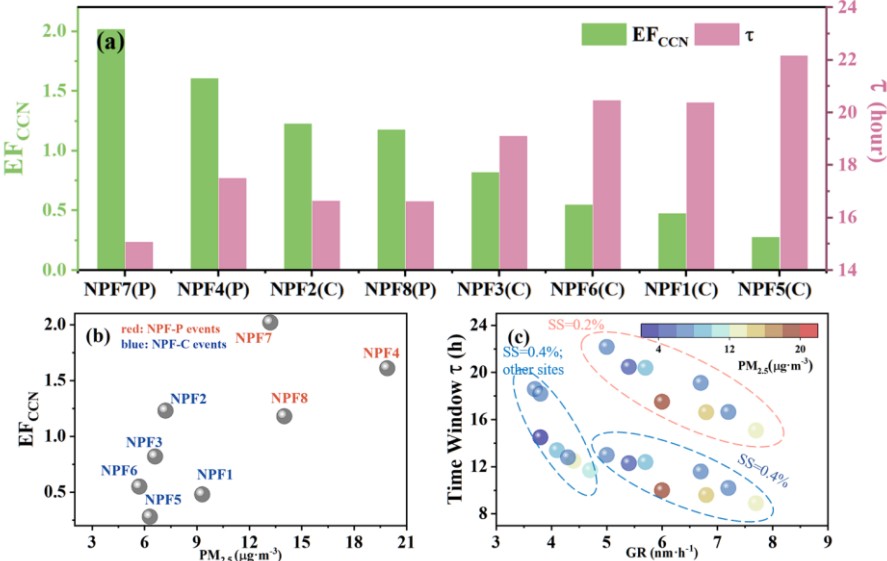


Figure 5: Relationships between CCN enhancement factors, Time Window (τ), and environmental
parameters. (a) Scatter plot of the CCN enhancement factor ($EF_{CCN}$) versus the Time Window (τ) for
particle growth to CCN size across all eight NPF events. (b) Correlation between the $PM_{2.5}$ mass
concentration and $EF_{CCN}$, where individual data points are color-coded to distinguish between NPF
events occurring under polluted (red font) and clean (blue font) conditions. (c) Relationship between the
particle growth rate (GR) and τ. The color gradient represents the concurrent $PM_{2.5}$ mass concentration
at the Shanghuang station for data at supersaturations of 0.2% and 0.4%. Data from other sites (shown
for SS=0.4%) are included for comparison.
Table 1: Summary of NPF events. For each event, the table lists the date, event type classification
(NPF-C/NPF-P), start time, average formation rate at 2.5 nm ($J_{2.5}$), average growth rate (GR),
condensation sink (CS), sulfuric acid (SA) concentration, key meteorological parameters (temperature,
T; relative humidity, RH; wind speed, WS), and the average number concentrations of nucleation(NUC),
Aitken(AIT), and accumulation(ACC) mode particles.

| | date | type | start time | $J_{2.5}$ (cm$^{-3}$ s$^{-1}$) | GR (nm h$^{-1}$) | CS (s$^{-1}$) | SA (cm$^{-3}$) | T (℃) | RH (%) | WS (m s$^{-1}$) | NUC (cm$^{-3}$) | AIT (cm$^{-3}$) | ACC (cm$^{-3}$) |
|---|---|---|---|---|---|---|---|---|---|---|---|---|---|
| **NPF-1** | 2024/4/28 | C | 9:00 | 0.6 | 4.8 | 0.007 | 6.1E+6 | 19.1 | 87 | 2.2 | 305 | 766 | 741 |
| **NPF-2** | 2024/5/5 | C | 6:00 | 0.8 | 5.7 | 0.004 | 6.2E+6 | 16.0 | 90 | 2.1 | 985 | 1552 | 304 |
| **NPF-3** | 2024/5/6 | C | 7:00 | 1.3 | 6.7 | 0.006 | 1.0E+7 | 20.2 | 65 | 2.6 | 3229 | 3105 | 554 |
| **NPF-4** | 2024/5/13 | P | 7:00 | 3.4 | 6.0 | 0.015 | 1.0E+7 | 16.2 | 47 | 1.7 | 1771 | 5231 | 1330 |
| **NPF-5** | 2024/5/17 | C | 9:00 | 0.3 | 5.0 | 0.014 | 5.5E+6 | 25.2 | 68 | 1.4 | 382 | 1835 | 1920 |
| **NPF-6** | 2024/5/26 | C | 8:00 | 0.7 | 5.4 | 0.007 | 6.3E+6 | 26.6 | 69 | 1.5 | 482 | 2476 | 644 |
| **NPF-7** | 2024/5/28 | P | 7:00 | 1.3 | 7.7 | 0.011 | 7.1E+6 | 17.6 | 60 | 2.2 | 522 | 2706 | 1073 |
| **NPF-8** | 2024/5/29 | P | 6:00 | 2.4 | 6.8 | 0.014 | 8.2E+6 | 22.5 | 43 | 1.6 | 1399 | 3123 | 1424 |



**Author contributions.**

ZR and WB designed the experiments, and WB, SS, JQ, YF, ZZ, RL, KY, GQ, XP, XL, LZ, WQ, YL, ZF and HB carried out the field measurements and data analysis. ZR performed the MALTE-BOX model simulation. WB and ZR interpreted the data and wrote the paper. All the authors contributed to discussing results and commenting on the paper.

**Competing interests.**

The authors declare that they have no known competing financial interests or personal relationships that could have appeared to influence the work reported in this paper.

**Financial support.**

This work was supported by the Strategic Priority Research Program of the Chinese Academy of Sciences (Grant No. XDB0760200), the National Natural Science Foundation of China [grant numbers 42275120, 42075111, and 42330605] and the National Key Research and Development Program [grant number 2023YFC3706101].

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
