# Peer review of "Oxidation-driven acceleration of NPF-to-CCN conversion"

_EGUsphere, 2025_

## Referee Comment (RC2)

**Review report on**

**Oxidation-driven acceleration of NPF-to-CCN conversion under polluted atmosphere: Evidence from mountain-top observations in Yangtze River Delta**

By Zhu, W., Shang, S., Wang, J., Wu, Y., Deng, Z., Ran, L., Kuang, Y., Tang, G., Huang, X., Pan, X., Liu, L., Xu, W., Sun, Y., Hu, B., Wang, Z., and Liu, Z.

Zhu et al.'s manuscript examined the new particle formation (NPF) mechanism at the top of the boundary layer, as this remains poorly understood in polluted environments. To this end, they took measurements at a mountaintop site in southeastern China, characterized NPF under different air masses, and assessed its contribution to cloud condensation nuclei (CCN). The authors identified 8 NPF events, during which significantly higher rates of particle formation and growth were observed in polluted conditions, driven by elevated levels of sulfuric acid and ammonia. Polluted air masses produced higher CCN enhancement and faster NPF-to-CCN conversion, which was accelerated by nitrate-induced particle growth. Their results show that, air masses influenced by pollution intensify and accelerate CCN production at the top of the boundary layer through enhanced atmospheric oxidation capacity.

However, several details are missing, and a more thorough discussion is required in specific sections. The paper is poorly written and too complex for readers to study, particularly part 3.4. Furthermore, the authors should use a revised version in better English. That version should be rejected. Other than that, the paper can be recommended for publication once the major issues listed below have been addressed. This will enhance the manuscript.

**Abstract**

**L20:** "... exploring the nucleation mechanism ..." --> Do the authors solely examine the nucleation mechanism throughout the manuscript? This means that they only consider the formation of the initial clusters and not the growth process. Please rephrase.

**L25-26:** According to the authors, ammonia generally enhances the nucleation process of sulfuric acid alone. This is vague. Please rephrase.

**L31-32:** Does nitrate have a significant effect on cloud formation? The authors are discussing the impact on CCN. While the CDNCs are indeed related to the CCN budget, this relationship is not straightforward. Please revise.

**The Abstract should be rewritten.**

**Introduction**

**L40-42:** Please rephrase. Furthermore, ensure that you use important references in your scientific sentences (e.g. Kulmala et al., 2001; Kerminen et al., 2018).

**L42-46:** Studies that report the significant impact of the NPF mechanism on CCN alone should also be included (e.g. Laaksonen et al., 2005; Kalkavouras et al., 2017; Kalkavouras et al., 2019).

**L52-54:** A reference is needed to confirm the negative impact of CS on the formation of nanoparticles (e.g. Kalivitis et al., 2019).

**L56-57:** However, many studies worldwide have used long-term measurements to demonstrate the role of NPF on CCN. This study only uses data from 8 NPF days. Please be more cautious.

**L59-61:** Please highlight the regional character of the NPF mechanism by including references from Aktypis et al. (2024) and Kalkavouras et al. (2021).

**L67-70:** It is vague. Which analysis is limited? Please rephrase and provide references.

L75-76: The authors argue that studies avoid determining the growth rate of small particles to CCN. However, it is quite common for this "growth speed" to be related to the start of the NPF and the time at which the CCN "feel" the NPF. Perhaps this sentence could be revised.

L76-79: Please emphasize the impact of anthropogenic pollutants on the rate of condensational growth by including the relevant references (e.g. Kalkavouras et al. 2020; Dinoi et al., 2023).

**L84-86:** A reference is needed.

L88-92: References are also needed.

**L103:** "... NPF nucleation and growth processes ... " → What does NPF nucleation mean? The NPF mechanism consists of atmospheric nucleation and the gradual growth of freshly formed particles. Therefore, this part of the sentence is incorrect. Please revise it.

L106-107: Could you please explain what "cloud processes" means?

**L107-108:** "... particle number size distributions (PNSD, 2nm~20  $\mu$ m), ..."  $\rightarrow$  There is probably a typo here (i.e. "~").

The Introduction should be revised. It lacks references, and better English should be used.

**Methodology**

**L116-120:** Please provide a map showing the exact location of the study area, even in the supplementary material. It would be useful for readers to have an idea of the location and to see pictures of your station. If possible, could you also add some references about this location? It is probably not the first time that a campaign has taken place there.

**L126-131:** There is a lot of missing information regarding the PNSD measurements. How many size bins does each instrument have? What are the aerosol and sheath flows? Was any calibration performed prior to the campaign? Please provide significant information about the quality of the experimental measurements.

**L142-145:** Please see the previous comment. The authors should provide more detail regarding the instrumentation setup. More information about ACSM measurements and data analysis should be provided: Standard/capture vapourizer? Did you apply any collection efficiency correction?

**L149-150:** Which instruments were calibrated? Did the authors mean the analyzers for the standard gaseous species (SO2, O3, NOx) and ammonia?

**L153:** This information should perhaps be included alongside the sentences on gaseous pollutants, rather than after the PM2.5.

**L219-223:** Firstly, the authors should avoid terms such as "mainstream" metric. Moreover, what exactly is the innovative approach, when using only 8 NPF episodes? According to Kalkavouras et al. (2019), when a **7-year dataset is used (162 NPF events were analyzed)**, the period from the start of NPF until the "wave" of new particles activated into CCN-relevant sizes is expressed through  $t_{start}$  and  $t_{decoupling}$ . Therefore, what valuable information is provided here? Furthermore, I find the hygroscopic growth confusing. The critical diameter is derived from kappa, however the authors state that  $\kappa$  is constantly changing. It is unclear.

The Methodology section definitely needs revising, as it is missing significant information.

**Results**

**L231-233:** There is a repetition here. See lines 164–167.

**L233:** Could you please provide the dates of these events? Given that this information is missing, it is likely that the authors mean typical Class I NPF events.

**L235:** What do "nucleation-mode particles" mean? The authors should provide all the relevant information in the methodology section. They should also explain what nucleation, Aitken and accumulation-mode particles are, and how they are calculated. Furthermore, please use a frame to present the 8 NPF events on the contour plot (Fig. 1), noting the dates of each event. The frame should include all the information, i.e. extend it to Fig. 1d, 1e and 1f.

**L241-243:** It would be helpful to provide a table containing all the NPF-related information. This table should show the dates, NPF frequency, the starting time, meteorological parameters on NPF days, the number concentrations of each particle mode, formation and growth rates, and so on, as well as a discussion of the information provided.

**L248-253:** "The data shows that he in-cloud formation of biogenic terpenoid"  $\rightarrow$  Something is missing. Moreover, to which data are the authors referring? See the previous comment. The discussion is rather complicated.

**L248-253:** "... significant variations in  $2\sim6$  nm Nucleation mode particles were observed among ..."  $\rightarrow$  The 2-6 nm size range belongs to the nucleation mode.

**L272:** In other words, does this mean that only particles in the 2–6 nm range belong to the nucleation mode? This is vague. Particles above 6 nm are considered to be in the Aitken mode. It is crucial that all this information is integrated into the Methods section.

Section 3.1 should be rewritten, as it is rather vague. The authors should present all the information in a clearer way, for example using tables, and the discussion should focus on this. Several gaps must be addressed.

**L279-280:** The authors began section 3.2 with the following sentence: "To further explore the chemical difference between NPF-P and NPF-C events, diurnal variation and average values of NPF parameters for NPF-C and NPF-P events were analyzed.". They then discuss the formation rate and the precursors (e.g.  $H_2SO_4$ ) that enhance it. Where exactly is the discussion of the chemical composition? Please be consistent throughout your manuscript. This seems quite complex.

L280-283: Poor English. Please rephrase.

**L283-284:** Are all three NPF-P events in peak at 10:00 LT? According to Fig. 2a, the time appears to be 12:00 LT.

**L289-290:** Are these the mean values of NH3? Please could you clarify and rephrase? Furthermore, the authors discuss the results from Fig. 2d. Following the discussion of H2SO4 in Fig. 2a, NH3 should be illustrated in Fig. 2b in the correct sequence.

**L291-292:** Do the values refer to the average? Moreover, there is a difference in  $O_3$  concentrations when NPF is taking place (10:00–12:00 LT), but it is not marked. The figure for  $O_3$  should be Fig. 2c, as discussed after ammonia.

**L293:** Could you please explain why "consequently" the growth rate is higher in NPF-P events than in NPF-C events? Please elaborate. Is there any reference to this outcome?

**L295:** The authors said: "Compared with European forested sites ...". However, they only used data from Hyytiälä in Finland. Please rephrase and use more references from forest and remote sites in China.

**L298-300:** "Collectively, the above results indicate that there are significant differences in the intensity of nucleation and growth processes of NPF events under different atmospheric conditions, and these differences are caused by different regional transport processes." → Please rephrase as: "These differences suggest that the intensity of an NPF event can vary significantly depending on the atmospheric conditions and the regional transport processes involved.". Atmospheric nucleation and subsequent growth are the NPF mechanism. Therefore, it is incorrect to refer to the "nucleation and growth processes of NPF events".

L302: Ammonia? The authors probably mean ammonium (NH4+).

**L301-302:** Here, the authors examine the role of chemistry in growth rates. How does the above statement that GR is "consequently" higher in NPF-P episodes hold up?

**L303:** To show the difference more clearly, please use the first y-axis for organics and the second y-axis for the other components.

**L303-304:** Is there any scientific explanation for this? Please provide a scientific discussion, rather than just presenting numbers.

**L307:** Was the ACSM used as a PM2.5 cyclone? Where is this information located in the manuscript? The SMPS recorded measurements in the size range of 2.5 nm to 16  $\mu$ m. The authors suggest that half of PM2.5 consists of organic matter. However, this size range differs from that on which the chemical analysis was based. Please elaborate.

L309: What are the latter stages of growth, and how do nitrates impact them?

**L310-311:** Provide some references. For instance, the comprehensive and holistic study by Trechera et al. (2023) revealed that the growth of nucleated particles is driven by the condensation of semi-volatile organic compounds.

**L311-313:** How did the authors reach this conclusion? Why are nitrates more active than organics? How was this outcome achieved?

**L314:** Since a CCN can be mainly activated at Aitken mode diameters, the focus will be on the chemical composition of PM1 rather than PM2.5. How scientifically sound is this approach?

Section 3.2 should be revised. There are many scientific omissions and errors in English. Furthermore, the figures presenting the diurnal variability of PM2.5, RH, SO2, NO2 and WS are not discussed at all.

**L319:** Provide a reference for the crucial role of H2SO4 in the NPF mechanism (e.g. Garcia Marlès et al. (2024)).

L320-324: However, the authors have already discussed H2SO4 in lines 284–289.

**L321:** Please provide comparisons with similar environments. Your station is not categorized as "urban".

**L323:** Please, see the previous comment.

**L323:** Which value remains significantly higher than those reported for clean sites? The *R*? Or is it something else? Please clarify.

L326-327: But why do the authors discussing the role of H2SO4 refer to ammonia and amines at this point?

**L331-332:** Please revise Figure 3a. The legend is captured with the data points. What is the  $J_{1.7}$  in the y-axis? There is no information about it in section 3.3. What does the "DMA" stand for? The authors should provide all the information.

**L345-346:** The authors have already discussed the scatter plot between J2.5 vs. H2SO4 in lines 321–324.

**L339-344:** The authors used the MALTE-BOX model to evaluate the formation mechanism in the presence of high levels of ammonia. But where is the discussion of these results? Why did they use this model when they had direct ammonia measurements during the campaign? In lines 345–351, they discuss the role of measured ammonia on NPF days. This discussion is vague.

Section 3.3 requires substantial scientific enhancement and a more detailed discussion.

**L354-355:** "To elucidate the relationship between the growth processes of the two types of NPF events and the formation of CCN." → Something is missing here. Please be aware of this throughout the manuscript.

**L360:** What does "Da" mean? It is  $d_c$ ; it is referred to as "critical diameter" above at which point all particles can act as CCN. Please rephrase.

**L364-368:** What about the condensation sink (CS)? The authors neither discuss nor calculate this decisive metric in the NPF mechanism. CS calculations should be performed for the entire study period and the results discussed.

L370: What does "TD" mean?

**L368-369:** "HNO3 enhances low-volatility organic compound production, further suppressing the hygroscopicity of NPF-P ultrafine particles"  $\rightarrow$  It is not clear to me. How does the enhancement of organic compounds suppress the hygroscopicity when the  $\kappa$  consists of organic and inorganic substances? Please could you elaborate?

**L373-374:** The authors write about figure 4b and then discuss figure 4d. This makes it difficult to follow the discussion.

**L374-375:** "During initial nucleation (0~2 hours), elevated non-volatile fractions (Figure 4d) suppress hygroscopic growth, maintaining  $D_a$  at higher levels (~120 nm)."  $\rightarrow$  This is difficult to follow. Where does this outcome stem from? It is quite vague.

L375-377: Where does this outcome come from, and how is it depicted?

**L377-380:** This makes it difficult to follow the discussion. I am unable to see all this information on the figures.

**L380:** Could you please add a tint to show the NPF days? It is difficult to examine the figures 4a, 4b, and 4c as they are.

L386-389: What are the main differences when compared to Figures 2h and 2i?

**L390-393:** How is this outcome supported? It seems to be a general conclusion that lacks scientific argumentation.

**L428-431:** I cannot understand what is being said. What and where is the role of nitrate?

Section 3.4 should be completely revised. It is unclear and difficult to read. It needs to be improved scientifically.

**References**

Aktypis et al. (2024), https://doi.org/10.5194/acp-24-65-2024

Dinoi et al. (2023), https://doi.org/10.5194/acp-23-2167-2023

Garcia Marlès et al. (2024), https://doi.org/10.1016/j.envint.2024.109149

Kalivitis et al. (2019), https://doi.org/10.5194/acp-19-2671-2019

Kalkavouras et al. (2017), https://doi.org/10.5194/acp-17-175-2017

Kalkavouras et al. (2020), https://doi.org/10.1016/j.atmosres.2020.104911

Kalkavouras et al. (2021), https://doi.org/10.3390/atmos12010013

Kerminen et al. (2018), DOI 10.1088/1748-9326/aadf3c

Kulmala et al. (2001), https://doi.org/10.3402/tellusb.v53i4.16622

Laaksonen et al. (2005), doi:10.1029/2004GL022092

Trechera et al. (2023), https://doi.org/10.1016/j.envint.2023.107744

---

## Author Comment (AC1)

**Anonymous Referee #1:** This manuscript investigates CCN production associated with atmospheric new particle formation using a few observed cases at a mountain-top measurement station. The scientific approach appears to be robust, and the paper has some novel features, such as introduction of the "time window" concept. Interpretation of the results requires, however, considerable revisions here and there. The language of the paper requires also improvements. My detailed comments in this regard are given below.

[*Response*] The authors sincerely thank the reviewer for the positive and constructive assessment, as well as for the thoughtful and thorough review. We appreciate the reviewer's recognition of the study's robust approach and novel features. In direct response to the general suggestion regarding language, the English throughout the entire manuscript has been thoroughly polished by a native-speaking expert to enhance clarity and readability. A detailed, point-by-point response to each specific comment follows below (in blue color), along with a description of the corresponding revisions made to the manuscript text (in red color). We hope that the revised manuscript, together with our clarifications, fully addresses all the concerns raised.

**Scientific issues**

**Lines 29-32:** in Abstract: While changes in CCN concentrations influence many cloud properties via changes in cloud droplet concentrations, the ability of a cloud to form is not really dependent on CCN concentrations (unless CCN are missing altogether, which is extremely rarely the case). I recommend mentioning changes in cloud properties in this context, not talking about cloud formation.

[*Response*] We thank the reviewer for the suggestion, and the mentioned sentence has been revised as:

"Furthermore, the duration of NPF-to-CCN conversion was quantified using a 'Time Window ($\tau$)', revealing that polluted conditions accelerated the conversion by 17.0% ($\tau$ = 16.4 h vs. 19.8 h). Nitrate played an important role in maintaining a rapid particle growth rate, thereby shortening $\tau$ and enhancing CCN production from NPF—a process that can ultimately influence cloud microphysical properties by increasing the potential cloud droplet number concentration."

**Lines 218-219:** Talking about hygroscopic growth in this context is misleading. Being a CCN at a given supersaturation (SS) is an aerosol property, dictated by aerosol size and chemical composition. Becoming a CCN means increasing either particle "dry" size (by condensation growth) or its hygrospicity to a sufficient degree. This should not be mixed with particle hygroscopic growth which happens when CCN respond to ambient RH or are activating into cloud droplets.

[*Response*] We thank the reviewer for pointing out this critical clarification, and the mentioned sentence has been revised as:

"This approach more clearly describes the dynamic process in which newly formed particles

grow via condensation (increasing dry size and/or altering chemical composition) to the critical size and hygroscopicity required to act as CCN at defined supersaturation, and thus extends current methodologies by offering a more process-explicit framework to evaluate how precursor conditions and chemical pathways modulate the climatic impact of NPF."

**Lines 311-313:** The logic of this statement does not work. The growth rate of a newly formed particle is determined solely by the condensation flux of low-volatile (and partly semi-volatile) compounds into it, being influenced mainly by the gas-phase concentration of such compounds. To a first approximation, the hygroscopicity of the condensing compounds do not matter in this context, as growth rates are usually determined for "dry" particles or for particles at low RH. It is true that the hygroscopicity of the condensing material influences the size of a growing particles at elevated RH, so that particles having more hygroscopic material have a higher "wet" size, which enhances the condensation flux and thereby growth rate of particles to some extent. But this a secondary effect compared with gas-phase concentrations.

[*Response*] We thank the reviewer for pointing out this important issue. We now clarify it as follow:

"Our findings indicate that in anthropogenically influenced mountain regions, nitrate— primarily as ammonium nitrate ($NH_4NO_3$)—can serve as a competitive source of low-volatility condensable vapor, partially substituting for organics in driving the mass growth of new particles. This occurs under conditions of elevated $NO_2$ and $NH_3$, where efficient photochemical production and gas-to-particle partitioning of $NH_4NO_3$ are favored. While the strong hygroscopicity of nitrate plays a secondary role by increasing the particle's wet size (and thus potentially enhancing condensation efficiency under high relative humidity), its primary contribution to growth is through direct vapor condensation."

**Lines 326-327:** Besides these theoretical calculations and laboratory experiments, there is also evidence form field measurements on the involvement of ammonia/amines. I recommend citing also field evidence in this context.

[*Response*] We thank the reviewer for the suggestion. The related references are cited here.

"Previous field and chamber studies also proposed that the gaseous species such as ammonia (Kulmala et al., 2013; Kürten et al., 2019) and amines (Metzger et al., 2010; Yao et al., 2018) also promote the nucleation."

**Line 359:** Was there some specific reason for why only a single SS (0.2%) was used in the calculations. Having two or even more values of SS in consideration may have given additional information on how newly formed particle reach CCN sizes. This is especially so when noting that with typical particle growth rates, it takes quite a while until newly formed particle grow into sizes relevant for CCN at 0.2% (at higher values of SS, which are certainly possible, newly formed particles become CCN much quicker).

[*Response*] We thank the reviewer for this insightful and constructive suggestion regarding

the consideration of multiple supersaturation (SS) levels. We agree with the reviewer that analyzing CCN activation at different SS values provides valuable information on the timescale and efficiency with which newly formed particles grow into CCN. Our initial focus on a single SS (0.2%) was primarily for clarity in presenting the core comparison between event types and for consistency with a referenced regional SS measurement (Gong et al., 2023). However, we fully agree that a multi-SS analysis significantly enriches the discussion. We now present the critical activation diameter ($D_a$) and the resultant CCN number concentration ($N_{CCN}$) for two representative SS values: 0.2% (representing a common in-cloud condition) and 0.4% (representing a higher, yet plausible, supersaturation). This dual-SS approach allows us to discuss how the required growth size and the subsequent CCN production potential vary with SS. The analysis confirms that at higher SS (0.4%), particles activate at a smaller $D_a$, leading to a shorter theoretical growth timescale ($\tau$) and a higher instantaneous $N_{CCN}$ from the growing nucleation mode. In addition, we have revised this part in section 3.4 accordingly.

"To quantify the CCN production from NPF events, the $N_{CCN}$ was calculated. Since supersaturation (SS) cannot be measured directly at the site, we employed a sensitivity approach using two representative SS values. These values were selected based on prior aircraft measurements in the regional background atmosphere, which reported a range of 0.1-0.5% (Gong et al., 2023). We performed calculations for SS=0.2% and SS=0.4%, encompassing a common in-cloud condition and a higher activation threshold. For each SS, the critical activation diameter ($D_a$) was derived using $\kappa$-Köhler theory, with the hygroscopicity parameter ($\kappa$) estimated from the measured particle chemical composition (Bougiatioti et al., 2011), adjusting for local altitude. The calculated $N_{CCN}$ for both SS levels was then compared with observed NCCN to evaluate the parameterization's performance and to analyze the SS-dependence of CCN production efficiency."

"The critical diameter for CCN activation ($D_a$) exhibited a strong dependence on supersaturation, as theoretically expected. For the studied NPF events, $D_a$ at SS=0.4% was substantially lower than at SS=0.2%. Under the lower SS condition (0.2%), $D_a$ varied from 111 to 129 nm, with a higher average in polluted (NPF-P) events (126 nm) compared to clean (NPF-C) events (120 nm). This difference correlated with a lower average hygroscopicity parameter ($\kappa$) for NPF-P events (0.18) than for NPF-C events (0.21), originating from a higher organic mass fraction (76.6% vs. 65.4%). At the higher SS of 0.4%, the average $D_a$ decreased to approximately 80 nm (NPF-P) and 76 nm (NPF-C), yet the inverse relationship between $D_a$ and $\kappa$ persisted.

The dependence of $D_a$ on supersaturation has significant implications for NPF-driven CCN production. At a higher SS of 0.4%, the substantially reduced critical diameter shortens the required growth trajectory and timescale, allowing particles to become CCN-active more

rapidly in environments with elevated supersaturation. Consequently, the net CCN enhancement ($EF_{CCN}$) during NPF was systematically greater at SS=0.4% than at 0.2%. Notably, while pollution-enhanced CCN production was evident at both SS levels, the relative enhancement of NPF-P over NPF-C events was more pronounced at the lower SS (0.2%). This indicates that the chemically processed, faster-growing particles in polluted air masses are particularly effective at overcoming the greater activation barrier (larger $D_a$) at low SS."

**Lines 362-364:** The logic of this statement does not work either. It is true that for 2 particles of similar size, the one with a higher organic fraction requires a higher SS to activate into a cloud droplet (as organics tend to have much lower hygroscopicity than the main inorganic compounds). However, a higher organic fraction is not expected to suppress condensation of other material from the gas phase (not even water unless these organics reduce e.g. the accommodation coefficients of condensing water). Please modify.

[*Response*] We thank the reviewer for pointing this out. We now modify it in the revised version according to your suggestion.

"This difference correlated with a lower average hygroscopicity parameter ($\kappa$) for NPF-P events (0.18) than for NPF-C events (0.21), originating from a higher organic mass fraction (77% vs. 65%)."

**Lines 390-393:** It is unclear to me how the degree of oxidation of organics can be estimated from VRF in mixtures of inorganic and organic compounds?

[*Response*] We thank the reviewer for pointing this out. We now provided more discussion in the revised version to clarify this issue.

"The VFR in the 14-80 nm size range was 10-20 % (Fig. 4d), significantly higher than values reported for polluted urban Beijing (~5 %; Wu et al., 2017). Because heating to 300 °C effectively removes volatile inorganic salts and semi-volatile organic compounds, a higher VFR primarily reflects a greater abundance of low-volatility organic compounds (LVOCs). At our background site, where local combustion influence is minimal, this points to a more aged, oxidized organic aerosol component (Ehn et al., 2014; Jimenez et al., 2009), consistent with the observed lower $\kappa$ and higher $D_a$."

**Lines 505-507:** There is something strange in this sentence. I am able to understand what is meant to be said.

[*Response*] We thank the reviewer for pointing this out. We now modify it in the revised version.

"Collectively, these results suggest that cross-regional pollutant transport enriches precursor concentrations, elevates the atmospheric oxidation capacity, and thereby enhances both the magnitude and advances the timing of CCN production at the boundary layer top. Crucially, while previous studies have indicated that intense local pollution can suppress CCN formation from NPF, our findings demonstrate that in oxidizing, transport-influenced environments such

as the one studied here, aged pollution plumes can instead amplify CCN yields. Accurately representing these oxidation-driven growth pathways in atmospheric models is therefore essential for constraining aerosol-cloud-climate feedbacks in rapidly developing regions."

**Technical issues**

1.  The paper has several typos, especially in section 3, which should be corrected in the revised version. Just a few examples of these: (line 231: event is, line 247: particles, line 248: the in-cloud, line 282: which is close to, line 296: GR values are higher by, lines 402-403: non-volatile)

    [*Response*] We sincerely thank the reviewer for this meticulous and helpful feedback. We have carefully addressed all the specific typographical errors and similar issues identified in Section 3 and throughout the manuscript. The detailed corrections are as follows:

    Line 231: Corrected "event are" with "event is"

    Line 247: Corrected "newly formed particle" with "newly formed particles".

    Line 248: The previous sentence has now been revised.

    Line 282: The sentence has been revised to: "Our values are close to those reported for other Chinese high-altitude background sites like Mount Tai ($J_3$= 1–2 cm$^{-3}$ s$^{-1}$; Shen et al., 2019) …"

    Line 296: The sentence has been revised to: "Compared to typical values reported for a remote boreal forest site (Hyytiälä, Finland: $J_3$= 0.4 cm$^{-3}$ s$^{-1}$, GR = 2.3 nm h$^{-1}$; Kerminen et al., 2018), the formation and growth rates observed at our site are higher by 275% and 126%, respectively."

    Lines 402-403: Corrected "no-volatile" with "non-volatile".

    We have conducted a thorough proofreading of the entire manuscript to eliminate any remaining typographical or grammatical errors.

2.  The percentages given in the paper appear overly accurate (one digit, e.g lines 289, 292 and 294, but also elsewhere). Maybe 1% accuracy would be more relevant here.

    [*Response*] We thank the reviewer for this insightful comment regarding the appropriate level of precision for reporting percentages. In response, we have systematically revised all quantitative results throughout the manuscript, including those on lines 289, 292, and 294, by rounding to the nearest whole number (e.g., from 23.2% to 23%, from 39.2% to 39%, from 23.6% to 24%).

3.  A similar accuracy issue concerns critical diameters in lines 360-362 (4 digits too many).

    [*Response*] We thank the reviewer's suggestion. The critical diameters are now reported

as integers, with all extraneous decimal places removed.

4. All the figures should be understandable based on available figure legends and figure captions, and the figures should be consistent with what is said in the text. This is not the case for many of the figures:

[*Response*] We sincerely thank the reviewer for highlighting this critical issue regarding the clarity and consistency of our figures. We have carefully reviewed each figure in the manuscript and implemented the comprehensive revisions to fully address your concern. First, we have thoroughly revised all figure captions to provide a complete, self-contained description of the data shown, including the specific conditions, parameters, and conclusions that can be drawn directly from the figure. All abbreviations and technical terms used in the figures are now clearly defined. Second, we have performed a line-by-line cross-check between the manuscript text and each figure. All descriptions, interpretations, and numerical references to the figures have been verified and, where necessary, corrected to ensure they accurately reflect the visual data presented.

5. In Figure 1, it is not mentioned which one (a or b) corresponds to clean or polluted conditions (I suppose a is clean and b is polluted based on the distributions).

[*Response*] We thank the reviewer for this precise and helpful feedback regarding Figure 1. We have revised the figure caption accordingly to explicitly state the conditions represented in each panel. The revised caption now reads:

[Figure]

Figure 1. Overview of atmospheric conditions and new particle formation (NPF) events at the mountain-top station. The dashed-line frame represents the NPF event days. (a-b) Lognormal-fitted particle number size distributions for representative (a) clean (NPF-C) and (b) polluted (NPF-P) NPF events. Fitted modes are color-coded: nucleation (<20nm, blue), Aitken (20-100nm, green), and accumulation (100-1000nm, orange). (c) Time series of observed particle number size distributions (dN/dlogDp) during the entire campaign. (d) Temporal evolution of particle types: cloud interstitial (dark red), cloud residual (light blue), and non-cloud periods (Ambient, light gray). The occurrence of sub-6nm particles (fresh nucleation) is overlaid as red lines, highlighting identified NPF event days. (e) Wind direction time series, where color intensity represents wind speed magnitude. (f) Time series of temperature and relative humidity.

6.  In Figure 2g, it is not explained which bars represent J and which ones GR (this info can now only be gotten by reading the main text).

    [*Response*] We thank the reviewer for this precise and helpful feedback regarding the clarity of Figure 2g. We have revised the figure legend and the corresponding figure element to explicitly differentiate the data series. The updated figure now clearly labels the bars representing the nucleation rate (J) and those representing the growth rate (GR), ensuring the information is directly accessible from the figure itself. In addition, we also revised the figure caption accordingly to explicitly state the conditions represented in each panel. The revised caption now reads:

[Figure]

Figure 2: Diurnal comparison of key parameters and new particle formation (NPF) metrics between clean (NPF-C) and polluted (NPF-P) event days. (a) formation rate ($J_{2.5}$); (b) $H_2SO_4$ concentration and condensation sink (CS); (c) $NH_3$ and $NO_2$ concentration (d) $O_3$ concentrations and UV-B radiation intensity; (e) $SO_2$ concentration and $PM_{2.5}$ mass concentration; (f) Temperature (T) and wind speed (WS); (g) Box plots of formation rate ($J_{2.5}$) and growth rate (GR), where boxes show the interquartile range (25th-75th percentile), internal lines denote the median, dots represent the arithmetic mean, and whiskers extend to the 10th and 90th percentiles. (h-i) Mean diurnal profiles of non-refractory $PM_{2.5}$ chemical composition (organics, sulfate, nitrate, ammonium, chloride) and black carbon (BC) mass concentration for (h) NPF-C and (i) NPF-P events.

7. The text says (lines 345-346) that Figs. 3b and 3c give J as a function of SA and ammonia product, but this is not true for Fig. 3b.

   [*Response*] We thank the reviewer for the careful correction. We have revised the relevant text in lines 345-346 to accurately reflect the content of the figures:

   "A pronounced linear relationship exists between $J_{2.5}$ and the product of $H_2SO_4$ and $NH_3$ concentrations (Fig. 3c). The Pearson correlation coefficient (R) for $J_{2.5}$ versus $[H_2SO_4]\times[NH_3]$ ranges from 0.79 to 0.92, notably higher than the correlation of $J_{2.5}$ with $[H_2SO_4]$ alone (R = 0.77-0.87, Fig.3b). "

8. The text says (lines 373-374) that Figs. 4b reveals something about the coupling between critical diameter and temporal particle size evolution. This is practically impossible to see from Figure 4b alone, but it requires additional information from other figures and text.

[*Response*] We sincerely thank the reviewer for this important observation. We admit that the coupling between critical diameter ($D_a$) and temporal particle size evolution cannot be directly discerned from Figure 4b alone, as this panel primarily presents the temporal evolution of $N_{CCN}$ and its activation ratio (AR). The coupling analysis is in fact based on the combined interpretation of Figure 4a (which shows the time-series of $D_a$) together with the particle size distribution data presented in Figure 1c and the growth rate information in Figure 2. We have therefore revised the text and added Figure S4 in Supplementary Information to clarify this issue.

"The diurnal evolution of the particle population further elucidates the transition from nucleation to CCN production. Total particle number concentration ($N_{CN}$) begins a rapid increase after ∼07:00 LT, driven by the nucleation burst (Fig. 4b). Although CCN concentration ($N_{CCN}$) starts to rise concurrently, the explosive production of small nucleation mode particles initially causes the activation ratio (AR = $N_{CCN}/N_{CN}$) to decline, reflecting the time required for growth to CCN active sizes. $N_{CCN}$ subsequently peaks around 09:00–10:00 LT, approximately 2–3 hours after the $N_{CN}$ surge, marking the period when a substantial fraction of newly formed particles has grown sufficiently. After ∼14:00 LT, as growth processes intensify (indicated by high GR), an increasing number of particles reach $D_a$, and the AR begins a gradual recovery (Figs. S4b-c)."

[Figure]

Figure S4: The diurnal variation of critical activation diameter ($D_a$), activation ratio (AR),

the number of cloud condensation nuclei ($N_{CCN}$) and total particle number concentration ($N_{CN}$) in NPF-C and NPF-P events. The blue line denotes to NPF-C events and red line denotes to NPF-P events.

9. In Figure 5b, the 3 red cases seem to refer to polluted ones, while the blue cases correspond to clean ones. This is not explained in figure caption, which is confusing especially as red and blue mean totally different things in Figure 5a.

   [*Response*] We sincerely thank the reviewer for this careful and constructive comment. We have now thoroughly revised the caption for Figure 5 to provide a complete and unambiguous description for each panel.

[Figure]

Figure 5: Relationships between CCN enhancement factors, Time Window ($\tau$), and environmental parameters. (a) Scatter plot of the CCN enhancement factor ($EF_{CCN}$) versus the Time Window ($\tau$) for particle growth to CCN size across all eight NPF events. (b) Correlation between the $PM_{2.5}$ mass concentration and $EF_{CCN}$, where individual data points are color-coded to distinguish between NPF events occurring under polluted (red font) and clean (blue font) conditions. (c) Relationship between the particle growth rate (GR) and $\tau$. The color gradient represents the concurrent $PM_{2.5}$ mass concentration at the Shanghuang station for data at supersaturations of 0.2% and 0.4%. Data from other sites (shown for SS=0.4%) are included for comparison.

**References**

Bougiatioti, A., Nenes, A., Fountoukis, C., Kalivitis, N., Pandis, S. N., & Mihalopoulos, N. (2011). Size-resolved CCN distributions and activation kinetics of aged continental and marine aerosol. Atmospheric Chemistry and Physics, 11(16), 8791-8808, https://doi.org/10.5194/acp-11-8791-2011.

Ehn, M., Thornton, J. A., Kleist, E., Sipilä, M., Junninen, H., Pullinen, I., ... & Mentel, T. F.

(2014). A large source of low-volatility secondary organic aerosol. Nature, 506(7489), 476-479, https://doi.org/10.1038/nature13032.

Gong, X., Wang, Y., Xie, H., Zhang, J., Lu, Z., Wood, R., ... & Wang, J. (2023). Maximum supersaturation in the marine boundary layer clouds over the North Atlantic. AGU advances, 4(6), e2022AV000855, https://doi.org/10.1029/2022AV000855.

Jimenez, J. L., Canagaratna, M. R., Donahue, N. M., Prevot, A. S. H., Zhang, Q., Kroll, J. H., ... & Worsnop, D. R. (2009). Evolution of organic aerosols in the atmosphere. science, 326(5959), 1525-1529, https://doi.org/10.1126/science.1180353.

Kulmala, M., Kontkanen, J., Junninen, H., Lehtipalo, K., Manninen, H. E., Nieminen, T., ... & Worsnop, D. R. (2013). Direct observations of atmospheric aerosol nucleation. Science, 339(6122), 943-946, https://doi.org/10.1126/science.1227385.

Kürten, A. (2019). New particle formation from sulfuric acid and ammonia: nucleation and growth model based on thermodynamics derived from CLOUD measurements for a wide range of conditions. Atmospheric chemistry and physics, 19(7), 5033-5050, https://doi.org/10.5194/acp-19-5033-2019.

Metzger, A., Verheggen, B., Dommen, J., Duplissy, J., Prevot, A. S., Weingartner, E., ... & Baltensperger, U. (2010). Evidence for the role of organics in aerosol particle formation under atmospheric conditions. Proceedings of the National Academy of Sciences, 107(15), 6646-6651, https://doi.org/10.1073/pnas.0911330107.

Wu, Z. J., Ma, N., Größ, J., Kecorius, S., Lu, K. D., Shang, D. J., ... & Zhang, Y. H. (2017). Thermodynamic properties of nanoparticles during new particle formation events in the atmosphere of North China Plain. Atmospheric research, 188, 55-63, https://doi.org/10.1016/j.atmosres.2017.01.007.

Yao, L., Garmash, O., Bianchi, F., Zheng, J., Yan, C., Kontkanen, J., ... & Wang, L. (2018). Atmospheric new particle formation from sulfuric acid and amines in a Chinese megacity. Science, 361(6399), 278-281, https://doi.org/10.1126/science.aao4839.

---

## Author Comment (AC2)

**Referee #2**:Zhu et al.'s manuscript examined the new particle formation (NPF) mechanism at the top of the boundary layer, as this remains poorly understood in polluted environments. To this end, they took measurements at a mountaintop site in southeastern China, characterized NPF under different air masses, and assessed its contribution to cloud condensation nuclei (CCN). The authors identified 8 NPF events, during which significantly higher rates of particle formation and growth were observed in polluted conditions, driven by elevated levels of sulfuric acid and ammonia. Polluted air masses produced higher CCN enhancement and faster NPF-to-CCN conversion, which was accelerated by nitrate-induced particle growth. Their results show that, air masses influenced by pollution intensify and accelerate CCN production at the top of the boundary layer through enhanced atmospheric oxidation capacity. However, several details are missing, and a more thorough discussion is required in specific sections. The paper is poorly written and too complex for readers to study, particularly part 3.4. Furthermore, the authors should use a revised version in better English. That version should be rejected. Other than that, the paper can be recommended for publication once the major issues listed below have been addressed. This will enhance the manuscript.

[*Response*] We sincerely thank the reviewer for their constructive and detailed evaluation of our manuscript. We are grateful for their recognition of the study's novel contributions regarding NPF mechanisms and CCN production at the boundary layer top under polluted conditions. We also deeply appreciate the reviewer's valuable suggestions for improvement. In direct response to the general concerns regarding language clarity and text complexity, we have undertaken the following comprehensive revisions. Now, the entire manuscript has been professionally edited by a native English speaker to enhance clarity, grammar, and overall readability. Specifically, Section 3.4, along with other technical sections identified as overly complex, has been substantially revised. We have simplified the narrative, clarified key scientific concepts, and restructured the paragraphs to improve logical flow and accessibility for a broad readership. In the following sections, we provide a detailed, point-by-point response (in blue color) to each of the reviewer's specific scientific and technical comments, and the changes in the manuscript are also provided (in red color). We hope that the revised manuscript, together with our clarifications, fully addresses all the concerns raised.

**Abstract**

**L20:** "… exploring the nucleation mechanism …" →Do the authors solely examine the nucleation mechanism throughout the manuscript? This means that they only consider the formation of the initial clusters and not the growth process. Please rephrase.

[*Response*] We sincerely thank the reviewer for the insightful comment. To avoid this potential ambiguity and to accurately reflect the full scope of our investigation, we have revised the sentence in the abstract as follows:

"Based on measurements at a mountain-top background site in southeastern China during spring

2024, this study systematically investigates the nucleation mechanism and subsequent growth dynamics of NPF events under contrasting air masses, and quantifies their role as a source of CCN."

**L25-26:** According to the authors, ammonia generally enhances the nucleation process of sulfuric acid alone. This is vague. Please rephrase.

[*Response*] We sincerely thank the reviewer for this valuable suggestion to clarify the role of ammonia. We admit that the original phrasing was somewhat vague. We have revised the sentence to incorporate both the observed correlation and the supporting theoretical evidence, as follows:

"The average formation rate ($J_{2.5}$: 2.4 vs. 0.7 $cm^{-3}\,s^{-1}$) and growth rate (GR: 6.8 vs. 5.5 $nm\,h^{-1}$) were significantly higher in NPF-P events than in NPF-C events, alongside elevated concentrations of sulfuric acid and ammonia. The correlation between log $J_3$ and [$H_2SO_4$], as well as theoretical simulations with the MALTE_BOX model, indicates that the enhanced nucleation in polluted conditions can be attributed to the participation of ammonia in stabilizing sulfuric acid-based clusters."

**L31-32:** Does nitrate have a significant effect on cloud formation? The authors are discussing the impact on CCN. While the CDNCs are indeed related to the CCN budget, this relationship is not straightforward. Please revise.

[*Response*] We thank the reviewer for raising this point, which was also concerned by Referee#1. We admit that the original phrasing oversimplified and inaccurately described the causal chain from CCN to cloud formation. We agree that the effect is on cloud microphysical properties (notably cloud droplet number concentration, CDNC), rather than on the fundamental ability of clouds to form. We have revised the relevant section to more accurately reflect this nuanced relationship:

"Furthermore, the duration of NPF-to-CCN conversion was quantified using a 'Time Window ($\tau$)', revealing that polluted conditions accelerated the conversion by 17.0% ($\tau$ =16.4 h vs. 19.8 h). Nitrate played an important role in maintaining a rapid particle growth rate, thereby shortening $\tau$ and enhancing CCN production from NPF-a process that can ultimately influence cloud microphysical properties by increasing the potential cloud droplet number concentration."

***The Abstract should be rewritten.***

[*Response*] Done, the Abstract was revised as follow:

"**Abstract.** To what extent the new particle formation (NPF) contributed to the cloud condensation nuclei (CCN) remained unclear, especially at the boundary layer top (BLT) in polluted atmosphere. Based on measurements at a mountain-top background site in southeastern China during spring 2024, this study systematically investigates the nucleation mechanism and subsequent growth dynamics of NPF events under contrasting air masses, and quantifies their role as a source of CCN. Eight NPF events were observed, and three of them occurred in the polluted conditions (NPF-P)

which associated with regional transportation while the rest five events appeared in the clean conditions (NPF-C). The average formation rate ($J_{2.5}$: 2.4 vs. 0.7 cm$^{-3}$ s$^{-1}$) and growth rate (GR: 6.8 vs. 5.5 nm h$^{-1}$) were significantly higher in NPF-P events than in NPF-C events, alongside elevated concentrations of sulfuric acid and ammonia. The correlation between log $J_3$ and [$H_2SO_4$], as well as theoretical simulations with the MALTE_BOX model, indicates that the enhanced nucleation in polluted conditions can be attributed to the participation of ammonia in stabilizing sulfuric acid-based clusters. In addition, much higher CCN enhancement factor was observed in NPF-P (EF$_{CCN}$: 1.6 vs. 0.7 in NPF-C) due to the regional transported of anthropogenic pollutants from the urban cluster regions and their secondary transformation under enhanced atmospheric oxidation capacity. Furthermore, the duration of NPF-to-CCN conversion was quantified using a 'Time Window ($\tau$)', revealing that polluted conditions accelerated the conversion by 17.0% ($\tau$ =16.4 h vs. 19.8 h). Nitrate played an important role in maintaining a rapid particle growth rate, thereby shortening $\tau$ and enhancing CCN production from NPF—a process that can ultimately influence cloud microphysical properties by increasing the potential cloud droplet number concentration. These findings reveal that polluted air masses enhance both the efficiency and speed of CCN production at the BLT through elevated atmospheric oxidation capacity."

**Introduction**

**L40-42:** Please rephrase. Furthermore, ensure that you use important references in your scientific sentences (e.g. Kulmala et al., 2001; Kerminen et al., 2018).

[*Response*] We thank the reviewer for the suggestion, and the mentioned references were added now.

"New Particle Formation (NPF) is the process in which low-volatility gaseous precursors nucleate to form stable nanoparticles, leading to rapid bursts in particle number concentration (Kulmala et al., 2001); these newly formed particles can subsequently grow to larger sizes via condensation of vapors or coagulation (Kerminen et al., 2018; Cai et al., 2024)."

**L42-46:** Studies that report the significant impact of the NPF mechanism on CCN alone should also be included (e.g. Laaksonen et al., 2005; Kalkavouras et al., 2017; Kalkavouras et al., 2019).

[*Response*] We thank the reviewer for the suggestion, and we have added the representative references.

"As an important source of atmospheric particles, NPF profoundly influences cloud microphysical properties, radiative forcing, and precipitation efficiency through its conversion process to CCN, thereby regulating regional and even global climate systems (Laaksonen et al., 2005; Kalkavouras et al., 2017; Kalkavouras et al., 2019)."

**L52-54:** A reference is needed to confirm the negative impact of CS on the formation of nanoparticles (e.g. Kalivitis et al., 2019).

[*Response*] We thank the reviewer for the suggestion, and the mentioned reference was added here.

"However, high condensation sinks (CS) also resulting from higher background particle concentrations strongly suppress nanoparticle formation intensity, accelerate scavenging of small particles, and may reduce particle hygroscopicity, thereby diminishing contribution of NPF to CCN (Kalivitis et al., 2019)."

**L56-57:** However, many studies worldwide have used long-term measurements to demonstrate the role of NPF on CCN. This study only uses data from 8 NPF days. Please be more cautious.

[Response] We sincerely thank the reviewer for this constructive critique, and agree that numerous long-term studies have indeed demonstrated the important contribution of NPF to CCN budgets globally. Our intention was not to overlook this well-established body of work, but to highlight that the specific chemical and dynamical pathways—particularly in understudied environments like high-altitude sites under complex pollution influence—are still not fully quantified. We agree that our study, based on a limited number of case events, cannot provide climatological statistics, but it offers detailed, process-level insights into the mechanisms that drive variability in CCN production efficiency. Following the reviewer's suggestion to be more cautious and precise, we have revised the sentence in the introduction to better reflect this nuance:

"Consequently, while numerous long-term observational studies have established the general importance of NPF as a source of CCN, the specific chemical pathways governing particle formation and subsequent growth into CCN under varying atmospheric conditions, particularly at high-altitude sites influenced by complex pollution regimes, remain inadequately constrained and require further validation through targeted observations."

**L59-61:** Please highlight the regional character of the NPF mechanism by including references from Aktypis et al. (2024) and Kalkavouras et al. (2021).

[*Response*] We thank the reviewer for the suggestion, and the mentioned references were added here.

"According to abundant field experiment observations, NPF typically manifests as "NPF events" within the global boundary layer; that is, the nucleation of nanoparticles and subsequent growth may occur over horizontal spatial scales extending up to tens or hundreds of kilometers, potentially with significant influence from anthropogenic emissions (Aktypis et al. 2024; Kalkavouras et al. 2021)."

**L67-70:** It is vague. Which analysis is limited? Please rephrase and provide references.

[Response]We thank the reviewer for raising this point and we understand the reviewer's concern. To make it clear, we revised this part and the related references were provided:

"Over the past three decades, the observational foundation for NPF has been substantially expanded, and numerous models have been developed to describe the process from both mechanistic and empirical perspectives. However, the contribution of NPF to the CCN budget

exhibits pronounced spatial heterogeneity. This variability stems largely from the high sensitivity of the subsequent particle growth process—through which newly formed particles evolve into CCN—to local environmental factors, including precursor chemical composition and growth mechanisms (Shen et al., 2016; Zhang et al., 2019). Consequently, despite advances in understanding NPF itself, constraints on the quantitative pathways from nucleation to CCN remain a significant source of uncertainty in aerosol-climate assessments (Kerminen et al., 2012)."

**L75-76:** The authors argue that studies avoid determining the growth rate of small particles to CCN. However, it is quite common for this "growth speed" to be related to the start of the NPF and the time at which the CCN "feel" the NPF. Perhaps this sentence could be revised.

[*Response*] We thank the reviewer for raising this point and we fully agree that our critique should focus on the directness and explicitness of the quantification, rather than the absence of any attempt to assess growth speed. This sentence was revised as follow:

"However, $EF_{CCN}$ primarily quantifies the net enhancement in CCN concentration resulting from an NPF event. While valuable for assessing the overall impact, this metric does not directly capture the kinetics of the underlying process, specifically, the rate at which the newly formed particle population grows to CCN-active sizes."

**L76-79:** Please emphasize the impact of anthropogenic pollutants on the rate of condensational growth by including the relevant references (e.g. Kalkavouras et al. 2020; Dinoi et al., 2023).

[*Response*] We thank the reviewer for the suggestion, and the mentioned references were added here.

"Anthropogenic pollutants in polluted atmospheres directly enhance the condensational growth rate of newly formed particles by increasing condensable vapor availability, as demonstrated in urban environments (Dinoi et al., 2023; Kalkavouras et al. 2020; Liu et al., 2018)."

**L84-86:** A reference is needed.

[*Response*] Thanks, done.

"China has emerged as a critical hotspot for studying NPF-to-CCN processes due to its dense urban clusters and complex interactions between anthropogenic and natural emissions. NPF events occur frequently in Chinese urban clusters (Chu et al., 2019), including the Yangtze River Delta (YRD)."

**L88-92:** References are also needed.

[*Response*] Thanks, done.

"The YRD area in China, as a globally representative region of intense anthropogenic emissions, provides abundant species for NPF nucleation and growth processes due to its high precursor concentrations ($SO_2$, $NH_3$, VOCs, etc.) and active photochemical oxidation processes (generating gaseous sulfuric acid, gaseous nitric acid, and secondary organic aerosols, among others) (Qi et al., 2018; Yao et al., 2018)"

**L103:** "… NPF nucleation and growth processes … "→What does NPF nucleation mean? The NPF mechanism consists of atmospheric nucleation and the gradual growth of freshly formed particles. Therefore, this part of the sentence is incorrect. Please revise it.

[*Response*] We thank the reviewer for the suggestion, and the sentence has been revised as follow:

"Therefore, it is critically important to elucidate how atmosphere with strong atmospheric oxidation capacity under polluted conditions at this BLT environment influence new particle formation and growth processes, ultimately determining the efficiency of their contribution to CCN production."

**L106-107:** Could you please explain what "cloud processes" means?

[*Response*] We thank the reviewer for the question. The phrase "cloud processes" was intended to broadly refer to cloud occurrence and related microphysical conditions (e.g., in-cloud scavenging, cloud droplet activation) typical at the high-altitude site during spring. However, as this point is not central to the main argument, we agree that removing it sharpens the sentence and avoids ambiguity.

"This study conducted comprehensive observations at a high-altitude BLT background site in YRD region in China during spring—a season characterized by frequent NPF events (Qi et al., 2015)."

**L107-108:** "… particle number size distributions (PNSD, 2nm~20 μm), …" →There is probably a typo here (i.e. "~").

[*Response*]Thanks. We have corrected it accordingly.

"By integrating data on particle number size distributions (PNSD, 2 nm-20 μm), aerosol chemical composition…"

***The Introduction should be revised. It lacks references, and better English should be used.***

[*Response*] We thank the reviewer for the suggestion and we have thoroughly revised the entire Introduction section. The revised version now includes more comprehensive and updated references to properly contextualize the study within the existing literature. Additionally, the English expression has been carefully polished by a native speaker to improve clarity, flow, and overall readability.

**Methodology**

**L116-120:** Please provide a map showing the exact location of the study area, even in the supplementary material. It would be useful for readers to have an idea of the location and to see pictures of your station. If possible, could you also add some references about this location? It is probably not the first time that a campaign has taken place there.

[*Response*] We thank the reviewer for the helpful suggestion to better describe the study site. A location map has been included as Supplementary Figure S1, clearly showing the geographical

position of the mountain-top station within the Yangtze River Delta region. And relevant references citing previous measurement campaigns and site characterization studies at this location have been added to the text.

[Figure]

Figure S1: Location of the Shanghuang station

"A continuous measurement was conducted at the Shanghuang Ecological and Environmental Observation of the Chinese Academy of Sciences (Shanghuang Station) from April 19 to May 30, 2024. This station is located at Mt. Damaojian (28.58°N,119.51°E, 1128 a.s.l) in Wuyi County, Zhejiang Province (Figure S1). It is characterized by mountainous terrain and forest coverage, representing a typical high-altitude background environment in the YRD region of China, more details about Shanghuang station could be found in Zhang et al. (2024) and Wang et al. (2025)."

**L126-131:** There is a lot of missing information regarding the PNSD measurements. How many size bins does each instrument have? What are the aerosol and sheath flows? Was any calibration performed prior to the campaign? Please provide significant information about the quality of the experimental measurements.

[*Response*] We thank the reviewer for the detailed comments regarding the description of our PNSD measurement system. We have thoroughly revised the Methods section (Section 2.1) to provide a more complete and rigorous account of the instrumentation, calibration procedures, and quality control.

"The particle number size distribution (PNSD) from 2.5 nm to 20 μm was continuously measured using an integrated system. The system consisted of a Neutral Cluster and Air Ion Spectrometer (NAIS, Airel Ltd.) covering a mobility diameter (dm) range of 2.5-42 nm, a scanning mobility particle sizer (SMPS, model 3936, TSI Inc.) for 14.5-710 nm (dm) comprising a model TSI3080 electrostatic classifier and a model TSI 3775 condensation particle counter, and an Aerodynamic

Particle Sizer (APS, model 3221, TSI Inc.) for 0.5-20 μm (aerodynamic diameter, da). Prior to and during the campaign, regular zero checks and flow-rate verifications were performed using a calibrated primary flow meter. The NAIS was operated at a sample flow rate of 60 L min$^{-1}$ to minimize diffusion losses, with data recorded at 10-min resolution (Mirme and Mirme, 2013). The SMPS was run with an aerosol-to-sheath flow ratio of 0.3:3.0 L min$^{-1}$ (1:10), and the APS with an aerosol flow of 1.0 L min$^{-1}$ and a sheath flow of 4.0 L min$^{-1}$ (Liu et al., 2016). Data from the SMPS and APS, recorded at 5-min resolution, were averaged into hourly spectra and merged into a unified particle size spectrum matrix (dm: 14.5 nm to 16,000 nm) following the procedure described by Beddows et al. (2010)."

**L142-145:** Please see the previous comment. The authors should provide more detail regarding the instrumentation setup. More information about ACSM measurements and data analysis should be provided: Standard/capture vapourizer? Did you apply any collection efficiency correction?

[*Response*] We thank the reviewer for the detailed comments regarding the description of our chemical composition measurements. We have thoroughly revised the Methods section (Section 2.1) to provide a more complete and rigorous account of the instrumentation, calibration procedures, and quality control.

"The chemical composition of non-refractory submicron particles (NR-PM$_{2.5}$), including organics, sulfate, nitrate, ammonium, and chloride, was measured using an Aerodyne Time-of-Flight Aerosol Chemical Speciation Monitor (ToF-ACSM, Li et al., 2023). The instrument sampled ambient air through the same inlet as the PNSD system, with a flow rate of 0.1 L min$^{-1}$ and a time resolution of 10 minutes. The ToF-ACSM was operated with a capture vaporizer, and its ionization efficiency (IE) was calibrated at the start of the campaign using 300 nm ammonium nitrate particles. The default relative ionization efficiencies (RIEs) for nitrate, organics, and chloride (1.1, 1.4, and 1.3, respectively) were applied (Nault et al., 2023). According to the ion efficiency (IE) calibration results using ammonium sulfate, the RIE values of ammonium and sulfate were 5.05 and 0.73, respectively (Zhang et al., 2024). A composition-dependent collection efficiency (CE) was applied to the raw data to correct for particle losses in the aerodynamic lens, following the parameterization established by Middlebrook et al. (2012)."

**L149-150:** Which instruments were calibrated? Did the authors mean the analyzers for the standard gaseous species (SO$_2$, O$_3$, NOx) and ammonia?

[*Response*] We thank the reviewer for this request for clarification. Yes, the sentence refers to the calibration of all gas-phase analyzers mentioned in the preceding text: specifically, the Thermo Scientific instruments for SO$_2$ (Model 43i), O$_3$ (Model 49i), and NO$_x$ (Model 42i), as well as the Picarro G1103 analyzer for NH$_3$. All these instruments were calibrated prior to the campaign to ensure measurement accuracy. We have revised the text to make this reference clearer.

"The concentrations of major gaseous precursors were measured using the following commercial analyzers: a pulsed UV fluorescence analyzer (Thermo Scientific, Model 43i) for sulfur dioxide

(SO$_2$), a UV photometric analyzer (Thermo Scientific, Model 49i) for ozone (O$_3$), a chemiluminescence analyzer (Thermo Scientific, Model 42i) for nitrogen oxides (NO$x$), and a cavity ring-down spectrometer (Picarro, Model G1103) for ammonia (NH$_3$). Prior to the campaign, all gaseous analyzers (SO$_2$, O$_3$, NO$x$, and NH$_3$) were calibrated with certified reference gases and zero air. In addition, routine calibration checks for these gaseous instruments were performed biweekly throughout the measurement period to ensure continuous accuracy and consistency of the data of gaseous pollutants."

**L153:** This information should perhaps be included alongside the sentences on gaseous pollutants, rather than after the PM$_{2.5}$.

[*Response*] We thank the reviewer for the suggestion, and we relocated the sentence and combined it with the sentences on gaseous pollutants.

**L219-223:** Firstly, the authors should avoid terms such as "mainstream" metric. Moreover, what exactly is the innovative approach, when using only 8 NPF episodes? According to Kalkavouras et al. (2019), when a 7-year dataset is used (162 NPF events were analyzed), the period from the start of NPF until the "wave" of new particles activated into CCN-relevant sizes is expressed through tstart and tdecoupling. Therefore, what valuable information is provided here? Furthermore, I find the hygroscopic growth confusing. The critical diameter is derived from kappa, however the authors state that κ is constantly changing. It is unclear.

[*Response*] We sincerely thank the reviewer for the insightful comment and for directing us to the valuable study by Kalkouras et al. (2019). We agree that the terminology "mainstream" is unnecessary and have removed it in the revised text.

The reviewer raises an important point regarding the innovation of our approach. The study by Kalkouras et al. (2019) provided a significant advancement by analyzing a 7-year dataset and conceptualizing the timescale from the nucleation burst to the "wave" of particle activation using the parameters $t_{\text{start}}$ and $t_{\text{decoupling}}$. Their work excellently captures the climatological feature of CCN production from NPF. Still, their method primarily defines the observational interval between the nucleation burst and the subsequent rise in CCN counts. This interval is a result of the combined effects of growth dynamics and varying background conditions, but it does not directly deconvolve or quantify the intrinsic, process-level kinetics of the growth path itself. Our proposed "Time Window (τ)" metric aims to address this specific gap. Instead of measuring the observed time lag, τ calculates the theoretical duration required for a particle to grow from a well-defined initial diameter (D$_0$) to the critical activation diameter (D$_a$). By directly linking the particle's growth rate (GR$_{\text{nuc}}$), its evolving hygroscopicity (via D$_a$, which is derived from a representative or size-resolved κ), and the target activation size, τ provides a process-oriented metric that isolates the efficiency of the physical growth step. This allows for a more mechanistic comparison of NPF events across different environments, independent of variations in background aerosol and nucleation start times.

While we acknowledge that our case study is based on 8 NPF events, the value of τ lies in its

general applicability as an analytical framework. To demonstrate this, we have applied the $\tau$ calculation not only to our dataset but also to published data from several other European background sites (e.g., Leipzig-TROPOS, Bösel, Melpitz, Hohenpeißenberg, and Zugspitze; ). The comparative analysis presented in Section 3.4 shows how $\tau$ effectively distinguishes the efficiency of NPF-to-CCN conversion under contrasting pollution regimes, providing a quantitative link between precursor conditions and CCN yield that is consistent across diverse sites. Therefore, our contribution is not merely the observation of 8 events but the introduction and cross-validation of a quantitative, process-based metric ($\tau$) that complements existing observational metrics like $EF_{CCN}$ or time-lag analyses.

Regarding the point on hygroscopic growth, we apologize for the lack of clarity. The critical diameter ($D_a$) is indeed calculated using $\kappa$-Köhler theory. During an NPF event, the average chemical composition (and thus $\kappa$) of the growing mode can evolve. In our calculation, we use a representative $\kappa$ value derived from the measured chemical composition of the growing mode during its evolution to CCN sizes, or a size-resolved $\kappa$ where available. This provides a best estimate of the effective hygroscopicity governing the activation step. We have revised it to explicitly clarify this point.

"The impact of NPF on CCN has been frequently assessed using metrics such as the CCN enhancement factor ($EF_{CCN}$) as mention in section 2.4.1. More recently, observational studies have conceptualized the timescale of this process by analyzing the interval between the nucleation burst and the subsequent increase in CCN concentration. For instance, Kalkouras et al. (2019) characterized this period through parameters such as $t_{start}$ and $t_{decoupling}$, which effectively capture the climatological time lag of CCN production from NPF events. Still, those methods do not directly deconvolve or quantify the intrinsic, process-level kinetics of the growth path itself. Building upon this foundation, the present study introduces a complementary, process-oriented metric—the "Time Window ($\tau$)"—to further quantify the intrinsic efficiency of CCN production during NPF. While metrics based on observational time lags reflect the net outcome influenced by both growth dynamics and variable background conditions, $\tau$ aims to isolate and quantify the core physical–chemical process: the theoretical time required for a newly formed particle to grow from its initial detectable diameter ($D_0$) to the critical activation diameter ($D_a$) at a given supersaturation. The activation diameter is derived from $\kappa$-Köhler theory, using an effective hygroscopicity parameter ($\kappa$) that represents the chemical composition of the growing nucleation mode. The time window $\tau$ (in hours) is calculated as:

$$\tau = (D_a - D_0)/GR_{nuc} \qquad (5)$$

where $GR_{nuc}$ is the observed growth rate of the nucleation mode. By directly linking the particle growth rate and its evolving hygroscopicity to the CCN activation threshold, $\tau$ provides a standardized, mechanistic measure that enables comparative analysis of NPF-to-CCN conversion efficiency across diverse atmospheric environments and pollution regimes. This approach more

clearly describes the dynamic process in which newly formed particles grow via condensation (increasing dry size and/or altering chemical composition) to the critical size and hygroscopicity required to act as CCN at defined supersaturation, and thus extends current methodologies by offering a more process-explicit framework to evaluate how precursor conditions and chemical pathways modulate the climatic impact of NPF."

***The Methodology section definitely needs revising, as it is missing significant information.***

[*Response*] We thank the reviewer for pointing out the need for a more comprehensive methodology section. We agree that a detailed and transparent description of the methods is essential. In response, we have thoroughly revised and expanded the entire Methodology section (Section 2).

**Results**

**L231-233:** There is a repetition here. See lines 164–167.

[*Response*] We thank the reviewer for reminding this, the repetition part was deleted and revised as follow:

"During the intense campaign, eight NPF events were identified across 39 valid observation days from April 19 to May 30 at the Shanghuang station."

**L233:** Could you please provide the dates of these events? Given that this information is missing, it is likely that the authors mean typical Class I NPF events.

[*Response*]We thank the reviewer for the suggestion, and a table (Table 1 as showed below) containing all the NPF-related information was added in the revised manuscript.

Table 1. Summary of NPF events. For each event, the table lists the date, event type classification (NPF-C/NPF-P), start time, average formation rate at 2.5 nm ($J_{2.5}$), average growth rate (GR), condensation sink (CS), sulfuric acid (SA) concentration, key meteorological parameters (temperature, T; relative humidity, RH; wind speed, WS), and the average number concentrations of nucleation(NUC), Aitken(AIT), and accumulation(ACC) mode particles.

| | date | type | start time | $J_{2.5}$ ($cm^{-3} s^{-1}$) | GR ($nm\ h^{-1}$) | CS ($s^{-1}$) | SA ($cm^{-3}$) | T (ºC) | RH (%) | WS ($m\ s^{-1}$) | NUC ($cm^{-3}$) | AIT ($cm^{-3}$) | ACC ($cm^{-3}$) |
|---|---|---|---|---|---|---|---|---|---|---|---|---|---|
| **NPF-1** | 2024/4/28 | C | 9:00 | 0.6 | 4.8 | 0.007 | 6.1E+6 | 19.1 | 87 | 2.2 | 305 | 766 | 741 |
| **NPF-2** | 2024/5/5 | C | 6:00 | 0.8 | 5.7 | 0.004 | 6.2E+6 | 16.0 | 90 | 2.1 | 985 | 1552 | 304 |
| **NPF-3** | 2024/5/6 | C | 7:00 | 1.3 | 6.7 | 0.006 | 1.0E+7 | 20.2 | 65 | 2.6 | 3229 | 3105 | 554 |
| **NPF-4** | 2024/5/13 | P | 7:00 | 3.4 | 6.0 | 0.015 | 1.0E+7 | 16.2 | 47 | 1.7 | 1771 | 5231 | 1330 |
| **NPF-5** | 2024/5/17 | C | 9:00 | 0.3 | 5.0 | 0.014 | 5.5E+6 | 25.2 | 68 | 1.4 | 382 | 1835 | 1920 |
| **NPF-6** | 2024/5/26 | C | 8:00 | 0.7 | 5.4 | 0.007 | 6.3E+6 | 26.6 | 69 | 1.5 | 482 | 2476 | 644 |
| **NPF-7** | 2024/5/28 | P | 7:00 | 1.3 | 7.7 | 0.011 | 7.1E+6 | 17.6 | 60 | 2.2 | 522 | 2706 | 1073 |
| **NPF-8** | 2024/5/29 | P | 6:00 | 2.4 | 6.8 | 0.014 | 8.2E+6 | 22.5 | 43 | 1.6 | 1399 | 3123 | 1424 |

**L235:** What do "nucleation-mode particles" mean? The authors should provide all the relevant information in the methodology section. They should also explain what nucleation, Aitken and accumulation-mode particles are, and how they are calculated. Furthermore, please use a frame to present the 8 NPF events on the contour plot (Fig. 1), noting the dates of each event. The frame should include all the information, i.e. extend it to Fig. 1d, 1e and 1f.

[*Response*] We thank the reviewer for these constructive suggestions regarding the clarity of particle mode definitions and the presentation of event information. First, we agree that the term "nucleation-mode particles" requires clarification in the methodology. We have added the information in Section 2.2 explaining the classification of particle size modes.

"Based on their size, atmospheric aerosol particles are commonly grouped into four modes: nucleation mode ($<20$ nm), Aitken mode (20-100 nm), accumulation mode (100-1000 nm), and coarse mode ($>1$ μm). In this study, the number concentration of each mode was obtained by integrating the measured particle number size distribution over the corresponding diameter interval. A NPF event is identified when a distinct and sustained ($\geq 2$ h) burst of nucleation-mode particles—particularly in the sub-6 nm size range—is observed, followed by a clear growth of the mode to larger sizes (Dal Maso et al., 2005)."

Second, we agree that marking the eight classified NPF events directly on Figure 1 would greatly enhance clarity. In the revised manuscript, we have added color-coded boxes in Figure 1, extending across panels c, d, e, and f, to clearly indicate the time periods of each of the eight NPF events. The corresponding dates are labeled on the figure. The two unclassified events (April 30 and May 16) are not highlighted with these boxes, visually distinguishing them from the analyzed events.

Finally, we have consistently replaced vague references to "nucleation-mode particle bursts" with the more precise description: "bursts in the concentration of freshly nucleated sub-6 nm particles.", and the revised sentences showed below:

"Note that bursts in the concentration of freshly nucleated sub-6 nm particles were also observed on April 30 and May 16 (see Figure 1d). However, these two episodes were not classified as NPF events because they occurred at night and were not followed by sustained growth of the nucleation mode to larger sizes, which is a key criterion for defining a full NPF event."

And the Figure 1 has been revised as:

[Figure]

Figure 1. Overview of atmospheric conditions and new particle formation (NPF) events at the mountain-top station. The dashed-line frame represents the NPF days. (a-b) Lognormal-fitted particle number size distributions for representative (a) clean (NPF-C) and (b) polluted (NPF-P) NPF events. Fitted modes are color-coded: nucleation (<20nm, blue), Aitken (20-100nm, green), and accumulation (100-1000nm, orange). (c) Time series of observed particle number size distributions (dN/dlogDp) during the entire campaign. (d) Temporal evolution of particle types: cloud interstitial (dark red), cloud residual (light blue), and non-cloud periods (Ambient, light gray). The occurrence of sub-6nm particles (fresh nucleation) is overlaid as red lines, highlighting identified NPF event days. (e) Wind direction time series, where color intensity represents wind speed magnitude. (f) Time series of temperature and relative humidity.

**L241-243:** It would be helpful to provide a table containing all the NPF-related information. This table should show the dates, NPF frequency, the starting time, meteorological parameters on NPF days, the number concentrations of each particle mode, formation and growth rates, and so on, as well as a discussion of the information provided.

[*Response*] We thank the reviewer for the suggestion, and a table (Table 1 as showed above) containing all the NPF-related information was added. In addition, discussions of the information in Table 1 were provided.

**L248-253:** "The data shows that he in-cloud formation of biogenic terpenoid" → Something is missing. Moreover, to which data are the authors referring? See the previous comment. The

discussion is rather complicated.

[*Response*] We thank the reviewer for pointing out the unclear phrasing and missing data reference in the original text. We have substantially revised this paragraph to address these concerns. In addition, we have relocated this revised discussion to the end of Section 3.1. This repositioning allows it to serve as a dedicated case study examining the potential influence of in-cloud aqueous-phase chemistry on new particle formation and growth.

"It is worth noting that the NPF event observed on May 5 (NPF-C) occurred during a cloud interstitial period under persistently high relative humidity ($> 90\%$), accompany with a slightly higher formation rate ($J_{2.5}=0.8$ cm$^{-3}$ s$^{-1}$) and growth rate (GR=5.7 nm h$^{-1}$) compared with the average value of the other NPF-C events (Table 1). We hypothesize that aqueous-phase chemical processes within the preceding cloud were pivotal. A mechanism analogous to the "post-fog growth" reported in the Arctic may be at play, whereby in-cloud reactions generate semi-volatile organic compounds (SVOCs) that later condense onto particles (Kecorius et al., 2023). While direct measurements of the specific SVOCs are not available, the elevated concentration of isoprene—a key biogenic precursor-on that day (0.3 ppbv compared to the 0.2 ppbv average for other NPF-C events) provides indirect support for enhanced biogenic activity and potential secondary organic aerosol formation pathways. Following cloud dissipation, these cloud-generated condensable vapors were released and, under sustained high humidity, rapidly condensed onto the newly formed nucleation-mode particles. This organic-dominated condensation likely surpassed the nitrate-driven growth observed in other events, facilitating sustained particle growth and enabling a larger fraction of the population to surpass the activation diameter and reach CCN sizes."

**L248-253:** "… significant variations in 2~6 nm Nucleation mode particles were observed among …" → The 2-6 nm size range belongs to the nucleation mode.

[*Response*] We thank the reviewer for this consistent and helpful comment regarding terminology. As noted in our previous response, we have revised the terminology throughout the manuscript to enhance clarity. Following this principle, the term "Nucleation mode particles" here has been replaced with the more specific descriptor "freshly nucleated sub-6 nm particles" which accurately refers to the initial cluster population and aligns with the event identification criteria discussed in the methodology. In addition, the legend label "Nucleation mode particle (2-6 nm)" in Figure 1d has been updated to "freshly nucleated sub-6 nm particles".

"As showed in Figure 1d, significant concentration variations in freshly nucleated sub-6 nm particles were observed among the eight NPF events, the peak value of which ranged from 246 to 1318 cm$^{-3}$."

**L272:** In other words, does this mean that only particles in the 2–6 nm range belong to the nucleation mode? This is vague. Particles above 6 nm are considered to be in the Aitken mode. It is crucial that all this information is integrated into the Methods section.

[*Response*] We thank the reviewer for this consistent and helpful comment regarding terminology, and following the suggestions mentioned before the sentence was revised as:

"The average PNSD during NPF-C events and NPF-P events were fitted as the sum of three mode lognormal distributions (Figures 1a-b, Hussein et al., 2005), and revealed that the Aitken mode particle concentrations in NPF-P events (3978 cm$^{-3}$) than NPF-C (1980 cm$^{-3}$), while the freshly nucleated sub-6 nm particles were lower in NPF-P events (575 cm$^{-3}$) than NPF-C (881 cm$^{-3}$)."

***Section 3.1 should be rewritten, as it is rather vague. The authors should present all the information in a clearer way, for example using tables, and the discussion should focus on this. Several gaps must be addressed.***

[*Response*]We thank the reviewer for the constructive feedback on Section 3.1. We agree that the original presentation could be clearer and more focused, and we have thoroughly restructured and rewritten Section 3.1.

**L279-280:** The authors began section 3.2 with the following sentence: "To further explore the chemical difference between NPF-P and NPF-C events, diurnal variation and average values of NPF parameters for NPF-C and NPF-P events were analyzed.". They then discuss the formation rate and the precursors (e.g. $H_2SO_4$) that enhance it. Where exactly is the discussion of the chemical composition? Please be consistent throughout your manuscript. This seems quite complex.

[*Response*] We thank the reviewer for pointing out the inconsistency between the section title and its initial content. We admit that the original opening sentence of Section 3.2 did not accurately introduce the comprehensive analysis presented in the section. The section does analyze chemical evolution, but its primary focus is the diurnal comparison of key parameters, including both chemical drivers (precursors, aerosol composition) and physical metrics (formation rate, growth rate), between the two event types. To address this, the title of Section 3.2 has been changed from "Chemical evolution of the NPF-C and NPF-P events" to a more precise and descriptive title: "Diurnal Comparison of Key Drivers and NPF Metrics between Clean and Polluted Events". In addition, we have rewritten the introductory text for this section to clearly state its scope:

"3.2 Diurnal Comparison of Key Drivers and NPF Metrics between Clean and Polluted Events

To elucidate the factors driving distinct NPF behaviors, this section presents a diurnal comparison of key parameters between clean (NPF-C) and polluted (NPF-P) event days."

**L280-283:** Poor English. Please rephrase.

[*Response*]Thanks, and the sentence has been revised as:

"As shown in Figure 2a, the average formation rate ($J_{2.5}$) during NPF-P events was 2.4 cm$^{-3}$ s$^{-1}$, approximately 3.6 times higher than during NPF-C events (0.7 cm$^{-3}$ s$^{-1}$).."

**L283-284:** Are all three NPF-P events in peak at 10:00 LT? According to Fig. 2a, the time appears to be 12:00 LT.

[*Response*] We thank the reviewer for this careful observation. We agree that the peak formation rate in Figure 2a appears around noon, not at 10:00 LT. Our original sentence was ambiguous. We intended to highlight that the largest relative difference (i.e., the greatest fold-increase) in $J_{2.5}$ between NPF-P and NPF-C events occurred at 10:00 LT, not that this was the absolute peak time. We have revised the text to clarify both the peak timings and the timing of the largest inter-event discrepancy.

"The peak $J_{2.5}$ in NPF-P events (6.2 cm$^{-3}$ s$^{-1}$ at 12:00 LT) was also higher and occurred one hour later than the peak in NPF-C events (1.8 cm$^{-3}$ s$^{-1}$ at 11:00 LT). The most pronounced enhancement, which showed a fivefold increase, was observed at 10:00 LT (2.5 vs. 0.5 cm$^{-3}$ s$^{-1}$)."

**L289-290:** Are these the mean values of NH$_3$? Please could you clarify and rephrase? Furthermore, the authors discuss the results from Fig. 2d. Following the discussion of H$_2$SO$_4$ in Fig. 2a, NH$_3$ should be illustrated in Fig. 2b in the correct sequence.

[*Response*] We thank the reviewer for these helpful points. Yes, the values given (8.1 ppbv vs. 4.1 ppbv) are the average NH$_3$ concentrations during NPF-P and NPF-C events, respectively. We have revised the text to explicitly state this. Following the comment, we have adjusted the narrative flow in Section 3.2. The discussion of H$_2$SO$_4$ (Figure 2b) is now followed by the discussion of NH$_3$ (Figure 2c) before moving to other parameters, ensuring a coherent order that matches the figure panels.

"The average NH$_3$ concentration during NPF-P events (8.1 ppbv) was approximately twice that during NPF-C events (4.1 ppbv; Figure 2c). This elevated NH$_3$ level, coinciding with higher H$_2$SO$_4$, likely contributed to the enhanced nucleation rates observed under polluted conditions by stabilizing sulfuric acid clusters."

[Figure]

Figure 2: Diurnal comparison of key parameters and NPF metrics between clean (NPF-C) and polluted (NPF-P) event days. (a) formation rate ($J_{2.5}$); (b) $H_2SO_4$ concentration and condensation sink (CS); (c) $NH_3$ and $NO_2$ concentration (d) $O_3$ concentrations and UV-B radiation intensity; (e) $SO_2$ concentration and $PM_{2.5}$ mass concentration; (f) Temperature (T) and wind speed (WS); (g) Box plots of formation rate ($J_{2.5}$) and growth rate (GR), where boxes show the interquartile range (25th-75th percentile), internal lines denote the median, dots represent the arithmetic mean, and whiskers extend to the 10th and 90th percentiles. (h-i) Mean diurnal profiles of non-refractory $PM_{2.5}$ chemical composition (organics, sulfate, nitrate, ammonium, chloride) and black carbon (BC) mass concentration for (h) NPF-C and (i) NPF-P events.

**L291-292:** Do the values refer to the average? Moreover, there is a difference in $O_3$ concentrations when NPF is taking place (10:00–12:00 LT), but it is not marked. The figure for $O_3$ should be Fig. 2c, as discussed after ammonia.

[*Response*] We thank the reviewer for the attentive comments. Yes, the values given (27.7 ppbv and 19.9 ppbv) are the average $O_3$ concentrations over the respective event periods (NPF-P and NPF-C). We have revised the text to explicitly state this. We agree that the relative difference during the core nucleation window (10:00-12:00 LT) is less pronounced than the full-event average. We have refined the statement to more accurately reflect the observed pattern. The revised text now acknowledges this nuance. As suggested, the discussion of $O_3$ now follows directly after $NH_3$, corresponding to the panel order in Figure 2 (Figure 2d).

"Concurrently, NPF-P events exhibited a higher event-average of background ozone ($O_3$) concentration (27.7 ppbv vs. 19.9 ppbv for NPF-C). Although the $O_3$ difference narrowed during the peak nucleation period (10:00-12:00 LT)—suggesting its primary role is in maintaining an enhanced oxidative environment conducive to precursor oxidation rather than directly driving the instantaneous nucleation burst—the difference expanded again after 15:00 LT, reaching a maximum in the late afternoon (18:00 LT; Figure 2d)."

**L293:** Could you please explain why "consequently" the growth rate is higher in NPF-P events than in NPF-C events? Please elaborate. Is there any reference to this outcome?

[*Response*] We thank the reviewer for the critical feedback on this point. We admit that the use of "consequently" was unjustified, as the previous sentence described a temporal pattern (difference expanding by 14:00 LT) but did not establish a mechanistic cause for the higher growth rate. The observed difference in growth rates is not a direct consequence of the ozone pattern discussed immediately before it. To address this, we have removed the causal link and revised the text to present the observed growth rate enhancement as a separate, key finding. We now explicitly link the higher growth rates under polluted conditions to the broader set of favorable factors analyzed in this section, such as elevated levels of condensable vapors (e.g., from nitrate and oxidized organics). This provides a clearer and more accurate explanation for the result.

"Although the $O_3$ difference narrowed during the peak nucleation period (10:00-12:00 LT)— suggesting its primary role is in maintaining an enhanced oxidative environment conducive to

precursor oxidation rather than directly driving the instantaneous nucleation burst—the difference expanded again after 15:00 LT, reaching a maximum in the late afternoon (18:00 LT; Figure 2d). This later period coincides with the sustained particle growth phase, where a stronger oxidative capacity likely facilitates the production of low-volatility condensable vapors, thereby influencing condensational growth. Correspondingly, the average particle growth rate (GR) during NPF-P events was $6.8\,\text{nm}\,\text{h}^{-1}$, which is 23.6% higher than during NPF-C events ($5.5\,\text{nm}\,\text{h}^{-1}$; Figure 2g). The overall elevated GR is consistent with a greater abundance of condensable vapors (e.g., nitrate and photochemically generated organics), which are discussed in the following sections."

**L295:** The authors said: "Compared with European forested sites …". However, they only used data from Hyytiälä in Finland. Please rephrase and use more references from forest and remote sites in China.

[*Response*] We thank the reviewer for the helpful suggestion to provide a more geographically balanced comparison. We agree that including data from a broader range of sites, especially within China, offers better context for our observations. Following this advice, we have revised the sentence to include a direct comparison with well-known high-altitude sites in China (Mount Tai (1534m), Shen et al., 2019; Mount Heng(1269m), Nie et al., 2014; Mount Yulong (3410m), Shang et al., 2018) alongside the reference to the European boreal forest site (Hyytiälä).

"Compared to typical values reported for a remote boreal forest site (Hyytiälä, Finland: $J_3$= $0.4\,\text{cm}^{-3}\,\text{s}^{-1}$, GR = $2.3\,\text{nm}\,\text{h}^{-1}$; Kerminen et al., 2018), the formation and growth rates observed at our site are higher by 275% and 126%, respectively. Our values are close to those reported for other Chinese high-altitude background sites like Mount Tai ($J_3$= $1\text{-}2\,\text{cm}^{-3}\,\text{s}^{-1}$; Shen et al., 2019), Mount Heng ($J_{15}$ = $0.15\text{-}0.45\,\text{cm}^{-3}\,\text{s}^{-1}$; Nie et al., 2014), and Mount Yulong ($J_3$ = $1.33\,\text{cm}^{-3}\,\text{s}^{-1}$; Shang et al., 2018)."

**L298-300:** "Collectively, the above results indicate that there are significant differences in the intensity of nucleation and growth processes of NPF events under different atmospheric conditions, and these differences are caused by different regional transport processes." →Please rephrase as: "These differences suggest that the intensity of an NPF event can vary significantly depending on the atmospheric conditions and the regional transport processes involved.". Atmospheric nucleation and subsequent growth are the NPF mechanism. Therefore, it is incorrect to refer to the "nucleation and growth processes of NPF events".

[*Response*] Thanks, and the sentence has been revised as:

"These differences suggest that the intensity of an NPF event can vary significantly depending on the atmospheric conditions and the regional transport processes involved."

**L302:** Ammonia? The authors probably mean ammonium (NH4+).

[*Response*] Thanks for pointing this typo and sorry for the mistake, corrected.

"To investigate the chemical differences driving nanoparticle growth during the two types of NPF events, the diurnal variations of chemical components (organics, sulfates, nitrates, ammonium,

chlorides, and black carbon) were analyzed during NPF evolution (Figures 2h-i)."

**L301-302:** Here, the authors examine the role of chemistry in growth rates. How does the above statement that GR is "consequently" higher in NPF-P episodes hold up?

[*Response*] We thank the reviewer for following up on this point. The concern raised here has already been addressed in our previous revisions.

**L303:** To show the difference more clearly, please use the first y-axis for organics and the second y-axis for the other components.

[*Response*] We thank the reviewer for the suggestion. Done.

**L303-304:** Is there any scientific explanation for this? Please provide a scientific discussion, rather than just presenting numbers.

[*Response*] We thank the reviewer for the constructive suggestion. We have substantially expanded the discussion to provide a scientific explanation for the more pronounced and sustained increases in organics and nitrates during NPF-P events. The revised text now explicitly links the observed higher precursor levels ($NO_2$, $NH_3$, $O_3$) and the enhanced oxidative environment to specific chemical pathways, as suggested in the reviewer's feedback.

"The results show that during NPF-P events, mass concentrations of all major chemical components increased alongside particle growth, with organics and nitrates exhibiting the most pronounced and sustained enhancement (Figures 2h-i). In contrast, NPF-C events displayed weaker and less persistent increases. While organics dominated the non-refractory $PM_{2.5}$ (NR-$PM_{2.5}$) mass fraction (accounting for more than half) during the growth phase in both event types, the chemical evolution pathways diverged significantly under anthropogenic influence. The stronger nitrate growth in NPF-P events can be attributed to a more favorable chemical environment. These events were characterized by significantly higher concentrations of $NO_2$ and $NH_3$ (Figure 2c). Photochemical modeling indicates that elevated $NO_2$ under stronger solar radiation leads to enhanced production of gaseous nitric acid ($HNO_3$) (Figure S3). In the presence of abundant $NH_3$, this $HNO_3$ efficiently partitions to the particle phase via neutralization, forming ammonium nitrate. This process explains the more than fivefold increase in nitrate peak concentrations during the later growth stages of NPF-P events, where nitrate became a key driver for sustained condensational growth.

Similarly, the more substantial organic mass increase during NPF-P events is linked to enhanced secondary organic aerosol (SOA) formation. Higher daytime $O_3$ concentrations (Figure 2d) suggest a more intense oxidative environment, which promotes the photochemical oxidation of volatile organic compounds (VOCs). Coupled with elevated ambient VOC levels (e.g., isoprene), this leads to the production of more low-volatility oxygenated organic molecules that readily condense onto growing particles. Therefore, the synergistic enhancement of nitrate and organic precursors under polluted, transport-influenced conditions provides a robust chemical explanation for the faster and more sustained particle growth observed during NPF-P events compared to NPF-C events."

**L307:** Was the ACSM used as a PM2.5 cyclone? Where is this information located in the manuscript? The SMPS recorded measurements in the size range of 2.5 nm to 16 μm. The authors suggest that half of PM2.5 consists of organic matter. However, this size range differs from that on which the chemical analysis was based. Please elaborate.

[*Response*] We thank the reviewer for raising these important technical points regarding instrument size cuts and data consistency. In our revised Methods section (Section 2.1), we now explicitly state that the ToF-ACSM sampled ambient air through the same inlet as the PNSD system. Note that it not used a single $PM_{2.5}$ cyclone, but using an advanced aerosol-cloud sampling inlet system, which alternated between the $PM_1$ cyclone, $PM_{2.5}$ cyclone and total suspended particulate (TSP) passage every 20min. This ensures that the chemical composition (NR-$PM_{2.5}$) and particle number size distribution data pertain to the same sampled aerosol population. The reviewer rightly notes a potential confusion between the SMPS range (up to 16 μm) and the ACSM measurement (NR-$PM_{2.5}$). Our statement regarding organics constituting "more than half" refers specifically to the non-refractory $PM_{2.5}$ mass fraction measured by the ACSM during the particle growth stage. We have revised the text to eliminate this ambiguity.

"While organics dominated the non-refractory $PM_{2.5}$ (NR-$PM_{2.5}$) mass fraction (accounting for more than half) during the growth phase in both event types, the chemical evolution pathways diverged significantly under anthropogenic influence."

**L309:** What are the latter stages of growth, and how do nitrates impact them?

[*Response*] We thank the reviewer for requesting clarification on the specific growth stages and the mechanistic role of nitrates. We have revised the relevant paragraph to address this. The "later growth stages" refer to the period after the initial nucleation burst (typically post-noon), when particles have grown beyond the nucleation mode (>20 nm) into the Aitken and early accumulation modes (50-100nm). This phase is critical for determining whether particles can reach CCN-active sizes. Our revised text now explicitly explains that the enhanced nitrate growth during these stages in NPF-P events.

"While organics dominated the non-refractory $PM_{2.5}$ (NR-$PM_{2.5}$) mass fraction (accounting for more than half) during the growth phase in both event types, the chemical evolution pathways diverged significantly under anthropogenic influence. The stronger nitrate growth in NPF-P events can be attributed to a more favorable chemical environment. These events were characterized by significantly higher concentrations of $NO_2$ and $NH_3$ (Figure 2c). Photochemical modeling indicates that elevated $NO_2$ under stronger solar radiation enhances the production of gaseous nitric acid ($HNO_3$) (Figure S3). In the presence of abundant $NH_3$, this $HNO_3$ efficiently partitions to the particle phase via neutralization, forming ammonium nitrate (Wang et al., 2022). This process explains the more than fivefold increase in nitrate peak concentrations during the later growth stages of NPF-P events, where nitrate became a key driver for sustained condensational growth."

**L310-311:** Provide some references. For instance, the comprehensive and holistic study by Trechera et al. (2023) revealed that the growth of nucleated particles is driven by the condensation of semi-volatile organic compounds.

[*Response*] We thank the reviewer for the suggestion, and we have incorporated this reference as suggested.

"Previous field studies have highlighted the importance of organics for new particle growth in remote regions (Pierce et al., 2012). Recent comprehensive analyses from multiple European cities further support this view, demonstrating that the growth of nucleated particles is often driven by the condensation of semi-volatile organic compounds (Trechera et al., 2023)."

**L311-313:** How did the authors reach this conclusion? Why are nitrates more active than organics? How was this outcome achieved?

[*Response*] We thank the reviewer for the insightful question regarding the specific role of nitrate, which was also concerned by the Referee #1. Our revised text clarifies this point by distinguishing between the two key properties of ammonium nitrate in this context. Firstly, under the high precursor concentrations ($NO_2$, $NH_3$) and oxidative conditions characteristic of polluted transport, ammonium nitrate acts as a low-volatility, condensable vapor, contributing to the mass flux driving particle growth. Secondly, once partitioned into the particle phase, its high hygroscopicity plays a secondary but complementary role: under sustained high humidity, it increases the particle's wet size, which can slightly enhance the condensation efficiency for other vapors. Therefore, the statement "nitrates partly substitute for organics" refers to the former mechanism—the supply of condensable mass—which can become competitive with or supplement organic condensation pathways under specific, nitrate-favorable chemical conditions, rather than implying a general superiority in reactivity. We have revised the sentence accordingly:

"Our findings indicate that in anthropogenically influenced mountain regions, nitrate— primarily as ammonium nitrate ($NH_4NO_3$)—can serve as a competitive source of low-volatility condensable vapor, partially substituting for organics in driving the mass growth of new particles. This occurs under conditions of elevated $NO_2$ and $NH_3$, where efficient photochemical production and gas-to-particle partitioning of $NH_4NO_3$ are favored. While the strong hygroscopicity of nitrate plays a secondary role by increasing the particle's wet size (and thus potentially enhancing condensation efficiency under high relative humidity), its primary contribution to growth is through direct vapor condensation."

**L314:** Since a CCN can be mainly activated at Aitken mode diameters, the focus will be on the chemical composition of $PM_1$ rather than $PM_{2.5}$. How scientifically sound is this approach?

[*Response*] We sincerely thank the reviewer for this critical and insightful question, which highlights an important methodological consideration. We agree that the CCN-active population primarily resides in the Aitken and smaller accumulation modes (approximately <200 nm), and ideally, the chemical composition of this specific size range should be directly measured. Our

reliance on the bulk PM$_{2.5}$ (non-refractory PM$_{2.5}$) composition from the ToF-ACSM is based on the following reasoning, which is well-supported in the literature for analyzing particle growth dynamics:

First, during a sustained nucleation and growth event, the condensing vapors are distributed across the entire growing aerosol population. Under conditions with minimal pre-existing accumulation mode particles (as is typical in a background mountain-top environment influenced by aged plumes rather than fresh primary emissions), the chemical composition measured for the bulk aerosol (PM$_{2.5}$) can be a reasonable proxy for the composition driving the growth of the nucleation and Aitken modes. This is because the mass increase observed by the ACSM during the event is predominantly due to the condensation of semi- and low-volatility vapors onto the growing particle population.

Second, this approach of using bulk submicron composition to infer the drivers of nanoparticle growth has been successfully applied in several key studies. Notably, Vakkari et al. (2015) explicitly validated this method. Their analysis demonstrated that during the daytime growth of nucleation mode particles, the changes in the bulk organic and sulfate mass concentrations were consistent with the estimated condensational requirements of the growing nanoparticles. This provided strong evidence that the bulk composition reflects the condensing species.

Nevertheless, we fully acknowledge that this is an approximation. As the reviewer implies, and as noted in other works (e.g., Ehn et al., 2014), the composition can vary with particle size, especially regarding the organic fraction's oxidation state and volatility. Therefore, while our current analysis using bulk PM$_{2.5}$ composition provides a robust and widely accepted first-order assessment of the dominant growth contributors, we agree that future studies would greatly benefit from size-resolved chemical measurements to directly quantify the condensing species onto the sub-100 nm population.

To make it clear, we revised these sentences as follow:

"It should be noted that the analysis of chemical drivers for particle growth in this study relies on the bulk non-refractory PM$_{2.5}$ (NR-PM$_{2.5}$) composition measured by the ToF-ACSM. While CCN activation at the studied supersaturations primarily involves particles in the Aitken and smaller accumulation modes (< 200 nm), we assert that the bulk PM$_{2.5}$ composition serves as a valid proxy for the condensing vapors during sustained NPF events under our background conditions. This is supported by the fact that during such events, the growth of the nucleation mode is the dominant source of new aerosol mass in the submicron range. Previous study indicates that changes in bulk organic and inorganic mass concentrations correlate well with the condensational needs of growing nanoparticles, making bulk composition a practical and informative metric for identifying dominant growth pathways (Vakkari et al., 2015). We acknowledge that size-dependent compositional differences may exist and represent an important avenue for future research with size-resolved instrumentation."

***Section 3.2 should be revised. There are many scientific omissions and errors in English.***

***Furthermore, the figures presenting the diurnal variability of PM2.5, RH, SO2, NO2 and WS are not discussed at all.***

[*Response*] We sincerely thank the reviewer for the comprehensive and detailed feedback on Section 3.2. We acknowledge that the original version contained scientific gaps, language issues, and insufficient discussion of key parameters presented in the figures. We have undertaken a thorough, point-by-point revision of Section 3.2 to address all the specific comments, as detailed in our individual responses above.

**L319:** Provide a reference for the crucial role of $H_2SO_4$ in the NPF mechanism (e.g. Garcia Marlès et al. (2024)).

[*Response*] We thank the reviewer for the suggestion, and the mentioned reference was added here.

"Gaseous sulfuric acid is recognized as an important specie in nucleation across NPF events (Gracia et al., 2024)."

**L320-324:** However, the authors have already discussed $H_2SO_4$ in lines 284–289.

[*Response*] We thank the reviewer for pointing out the potential overlap in the discussion of sulfuric acid ($H_2SO_4$) between sections. Now we have revised both sections to sharpen their respective focuses, as detailed below.

Revised Text for Section 3.2 (Previous Lines 284-289):

"To elucidate the factors driving distinct NPF behaviors, this section presents a diurnal comparison of key parameters between clean (NPF-C) and polluted (NPF-P) event days. As shown in Figure 2a, the average formation rate ($J_{2.5}$) during NPF-P events was 2.4 $cm^{-3}$ $s^{-1}$, approximately 3.6 times higher than during NPF-C events (0.7 $cm^{-3}$ $s^{-1}$). The peak $J_{2.5}$ in NPF-P events (6.2 $cm^{-3}$ $s^{-1}$ at 12:00 LT) was also higher and occurred one hour later than the peak in NPF-C events (1.8 $cm^{-3}$ $s^{-1}$ at 11:00 LT). The most pronounced enhancement, which showed a fivefold increase, was observed at 10:00 LT (2.5 vs. 0.5 $cm^{-3}$ $s^{-1}$). While the average gaseous sulfuric acid ($H_2SO_4$) concentration was 23 % higher in NPF-P events ($8.1 \times 10^6$ $cm^{-3}$) and the condensation sink (CS) was also elevated (0.013 vs. 0.008 $s^{-1}$ for NPF-C), the significantly stronger formation and growth rates indicate that enhanced production of condensable vapors from anthropogenic pollution was sufficient to overcome the increased sink strength, enabling intense NPF—a phenomenon documented in other polluted environments (Yang et al., 2021). Crucially, the 23 % difference in [$H_2SO_4$] alone cannot account for the ~3.6-fold difference in $J_{2.5}$."

Revised Text for Section 3.3 (previous Lines 320-326)

"The correlation coefficients (R) between $J_{2.5}$ and [$H_2SO_4$] were 0.77 for NPF-C events and 0.87 for NPF-P events (Figure 3b). This positive dependence of the nucleation rate on sulfuric acid concentration is consistent with observations from remote background sites, though the strength of the correlation varies with the degree of anthropogenic influence (Kulmala et al., 2013)"

**L321:** Please provide comparisons with similar environments. Your station is not categorized as "urban".

[*Response*] We thank the reviewer for pointing out it. As suggested by your previous comment, the mentioned sentence has been deleted here and relocated in Section 3.2.

**L323:** Please, see the previous comment.

[*Response*] Thank you for your suggestion. We have now added observational results from remote sites and the related discussion was revised and provided.

"At pristine sites such as Hyytiälä, the correlation is often moderated by the co-involvement of biogenic organic vapors and ions (Kulmala et al., 2025), whereas at background sites in China affected by regional pollution transport, stronger correlations between nucleation and [$H_2SO_4$] was typically observed (Gao et al., 2025)."

**L323:** Which value remains significantly higher than those reported for clean sites? The R? Or is it something else? Please clarify.

[*Response*] We thank the reviewer for pointing out the ambiguous phrasing. In the revised text, we have clarified that the correlation coefficients (R=0.77–0.87) observed at our site are significantly higher than the typical R values reported for boreal forest sites like Hyytiälä.

"The correlation coefficients (R) between $J_{2.5}$ and [$H_2SO_4$] were 0.77 for NPF-C events and 0.87 for NPF-P events (Figure 3b). This positive dependence of the nucleation rate on sulfuric acid concentration is consistent with observations from remote background sites, though the strength of the correlation varies with the degree of anthropogenic influence (Kulmala et al., 2013). At pristine sites such as Hyytiälä, the correlation is often moderated by the co-involvement of biogenic organic vapors and ions (Kulmala et al., 2025), whereas at background sites in China affected by regional pollution transport, stronger correlations between nucleation and [$H_2SO_4$] was typically observed (Gao et al., 2025)."

**L326-327:** But why do the authors discussing the role of H2SO4 refer to ammonia and amines at this point?

[*Response*] We thank the reviewer for highlighting the abrupt transition in the original text. To make it clear, we revised these sentences and provided more discussion here.

"The correlation coefficients (R) between $J_{2.5}$ and [$H_2SO_4$] were 0.77 for NPF-C events and 0.87 for NPF-P events (Figure 3b). This positive dependence of the nucleation rate on sulfuric acid concentration is consistent with observations from remote background sites, though the strength of the correlation varies with the degree of anthropogenic influence (Kulmala et al., 2013). At pristine sites such as Hyytiälä, the correlation is often moderated by the co-involvement of biogenic organic vapors and ions (Kulmala et al., 2025), whereas at background sites in China affected by regional pollution transport, stronger correlations between nucleation and [$H_2SO_4$] was typically observed (Gao et al., 2025). The high correlations observed here (R = 0.77–0.87) align with the latter pattern, reinforcing that our mountain-top station, although a background site, experiences substantial anthropogenic influence that shapes the nucleation mechanism. However,

the moderate difference in [$H_2SO_4$] alone cannot explain the large difference in $J_{2.5}$ between event types (Section 3.2). Previous studies have also indicated that binary $H_2SO_4$–$H_2O$ nucleation cannot fully account for atmospheric NPF rates (Kirkby et al., 2011). This points to the importance of additional compounds that stabilize $H_2SO_4$ clusters and modulate nucleation efficiency. In particular, basic gases such as ammonia ($NH_3$) and amines are known to significantly enhance sulfuric acid-driven nucleation, as demonstrated by both theoretical and observational work (e.g., Kürten et al., 2018; Metzger et al., 2010). The elevated $NH_3$ concentrations measured during NPF-P events (Figure 2b) thus provide a plausible explanation for their higher nucleation rates despite a less-than-proportional increase in [$H_2SO_4$]."

**L331-332:** Please revise Figure 3a. The legend is captured with the data points. What is the J1.7 in the y-axis? There is no information about it in section 3.3. What does the "DMA" stand for? The authors should provide all the information.

[*Response*] We thank the reviewer for the careful comments regarding Figure 3. We have revised the figure and its caption to provide all the requested information clearly.

"Figure 3: Nucleation mechanism analysis at Shanghuang station. (a) Comparison of formation rates as a function of $H_2SO_4$ concentration among field observations, CLOUD chamber experiments, and theoretical predictions. Field measurements are presented as the 2.5 nm formation rate ($J_{2.5}$; colored circles: hollow for NPF-C events, solid for NPF-P events). These are compared with the 1.7 nm formation rate ($J_{1.7}$; squares and triangles) from CLOUD experiments conducted at 278 K and 38% RH under controlled precursor conditions: $H_2SO_4$-$NH_3$-$H_2O$ ternary nucleation (squares, $NH_3$=0.1 ppbv and 1 ppbv) and $H_2SO_4$-DMA-$H_2O$ ion-mediated nucleation (triangles, DMA=13-140 pptv) (Kürten et al., 2019; Almeida et al., 2013). DMA denotes dimethylamine. Color gradients indicate $NH_3$ (blue) and DMA (red) mixing ratios in the chamber. The yellow line shows the MALTE-BOX model prediction for $H_2SO_4$ nucleation with 5 pptv $NH_3$; the gray band represents the uncertainty in cluster binding energy ($\pm 1$ kcal mol$^{-1}$). (b) Formation rates ($J_{2.5}$) versus $H_2SO_4$ concentration for NPF-C (black squares) and NPF-P (red hollow circles) events. (c) Formation rates ($J_{2.5}$) as a function of the $H_2SO_4$ and $NH_3$ concentration for NPF-C (black squares) and NPF-P (red hollow circles), with Pearson correlation coefficients (R) indicated."

**L345-346:** The authors have already discussed the scatter plot between $J_{2.5}$ vs. $H_2SO_4$ in lines 321–324.

[*Response*] We thank the reviewer for pointing out it, and we revised this sentence.

**L339-344:** The authors used the MALTE-BOX model to evaluate the formation mechanism in the presence of high levels of ammonia. But where is the discussion of these results? Why did they use this model when they had direct ammonia measurements during the campaign? In lines 345–351, they discuss the role of measured ammonia on NPF days. This discussion is vague.

[*Response*] We thank the reviewer for the insightful questions regarding our use of the

MALTE-BOX model and the discussion of ammonia's role. The reviewer asks why we used the MALTE-BOX model despite having direct $NH_3$ measurements. While our measurements quantify ambient $NH_3$ levels, they cannot by themselves diagnose the specific nucleation mechanism or quantify its efficiency under our field conditions. Direct comparison of our field-derived formation rates ($J_{2.5}$) with the CLOUD chamber results ($J_{1.7}$), which was usually performed in our previous studies (Yang et al., 2021) and other related studies (Yao et al., 2018), is challenging because the CLOUD experiments were conducted under controlled but simplified conditions (e.g., fixed temperature, RH, and precursor ratios) that differ from our variable ambient environment. The MALTE-BOX model, which integrates the Atmospheric Cluster Dynamics Code (ACDC), allows us to bridge this gap. By inputting the average atmospheric conditions (CS, T, RH, pressure) and a fixed, representative $NH_3$ concentration (5 ppbv) from our campaign, we can simulate the theoretical sulfuric acid nucleation rate as a function of $[H_2SO_4]$ for a mechanism consistent with our environment. This provides a process-level, theoretical baseline against which to compare our observations.

We admit that the original text lacked a discussion of the model results. We have now substantially expanded this section.

"To explore the nucleation mechanism in the atmospheric boundary layer top, the relationship between $J_{2.5}$ and $[H_2SO_4]$ was analyzed for NPF-P and NPF-C events and compared with results from CLOUD chamber experiments, which delineate pathways for $H_2SO_4$–$NH_3$–$H_2O$ and $H_2SO_4$–dimethylamine (DMA)–$H_2O$ nucleation (Kürten et al., 2019; Almeida et al., 2013). As shown in Figure 3a, our measured formation rates (solid circles: NPF-P; hollow circles: NPF-C) fall within the $[H_2SO_4]$ range spanned by these two mechanisms in the chamber. Achieving the observed $J_{2.5}$ would require either higher DMA levels or higher $NH_3$ concentrations than those set in the specific CLOUD runs. Given the lack of significant DMA sources in the region (e.g., textile or industrial activities; Chang et al., 2022), ambient $NH_3$ (average ~5 ppbv during NPF) is the more plausible stabilizing base. However, the CLOUD experiments have not yet performed under similar atmospheric conditions as our field observation (e.g. higher $NH_3$ levels exceed 1ppbv) (Kürten et al., 2019). Thus, to evaluate the formation mechanism under rich-$NH_3$ conditions representative of our site, we performed simulations using the MALTE-BOX model (Boy et al., 2006; McGrath et al., 2012), which couples the Atmospheric Cluster Dynamics Code (ACDC). Input parameters were set to the average conditions during NPF events: condensation sink (CS) = 0.010 $s^{-1}$, $[NH_3]$ = 5 ppbv, RH = 66%, T = 293 K, and pressure = 883 hPa. The model calculates the formation rate for clusters growing past a critical size as a function of $[H_2SO_4]$. The simulation results are shown as the yellow line and gray uncertainty band in Figure 3a. Most of our measured $J_{2.5}$ data points fall within or near the model-predicted band, indicating that $H_2SO_4$-$NH_3$ nucleation is a quantitatively plausible mechanism under the observed conditions. The model predictions tend to be slightly higher than the measured rates. This discrepancy may arise because the model's initial cluster definition (e.g., a $(H_2SO_4)_5(NH_3)_5$ cluster corresponding to ~1.07 nm; Huang et al., 2016)

effectively simulates formation at a smaller size than our observational threshold ($J_{2.5}$), and potential uncertainties in cluster binding energies or the omission of other stabilizing species (e.g., organic vapors) in the simulation. Nevertheless, the general agreement supports the conclusion that ammonia-enhanced sulfuric acid nucleation is a dominant pathway at this site.

Independent support for the role of ammonia comes from the field-observed correlations. A pronounced linear relationship exists between $J_{2.5}$ and the product of $H_2SO_4$ and $NH_3$ concentrations (Figure 3c). The Pearson correlation coefficient (R) for $J_{2.5}$ versus $[H_2SO_4] \times [NH_3]$ ranges from 0.79 to 0.92, notably higher than the correlation of $J_{2.5}$ with $[H_2SO_4]$ alone (R = 0.77-0.87). This enhanced correlation when $NH_3$ is included as a co-variable has been observed in other polluted environments; for example, wintertime measurements in Shanghai reported a tighter relationship between $J_{1.34}$ and $[NH_3]$ ($R^2=0.62$) than with $[H_2SO_4]$ ($R^2= 0.38$) (Xiao et al., 2015). Together, the consistency between our observations and the MALTE-BOX simulations, combined with the strong field-based correlation that explicitly includes $NH_3$, provides robust evidence that ammonia plays a key role in enhancing sulfuric acid-driven nucleation at this mountain-top site."

***Section 3.3 requires substantial scientific enhancement and a more detailed discussion.***

[*Response*] We sincerely thank the reviewer for the detailed and constructive feedback on Section 3.3. We fully agree that this section, which discusses the nucleation mechanism, is critical and required substantial strengthening. We have undertaken a comprehensive revision of Section 3.3 to address all the specific points raised and to enhance its overall scientific depth and clarity. We are grateful for the reviewer's thorough review, which has been instrumental in improving this key part of our study.

**L354-355:** "To elucidate the relationship between the growth processes of the two types of NPF events and the formation of CCN." → Something is missing here. Please be aware of this throughout the manuscript.

[*Response*] We thank the reviewer for pointing out the incomplete sentence. We have revised it to form a complete, declarative sentence that clearly states the objective of Section 3.4.

"This section aims to elucidate the relationship between the growth processes of the two types of NPF events and their efficiency in forming CCN."

**L360:** What does "Da" mean? It is *dc*; it is referred to as "critical diameter" above at which point all particles can act as CCN. Please rephrase.

[*Response*] We thank the reviewer for reminding this. We have revised this sentence as follow:

"The critical diameter for CCN activation ($D_a$) exhibited a strong dependence on supersaturation (SS) …"

**L364-368:** What about the condensation sink (CS)? The authors neither discuss nor calculate this decisive metric in the NPF mechanism. CS calculations should be performed for the entire study

period and the results discussed.

[*Response*] We sincerely thank the reviewer for this critical and insightful point. We agree that the condensation sink (CS) is a decisive parameter in NPF, and its omission from the discussion was a significant oversight. Our analysis confirms the reviewer's observation: the CS during NPF-P events (0.013 s$^{-1}$) was indeed higher than during NPF-C events (0.008 s$^{-1}$). A higher CS indicates stronger competition for condensable vapors by the pre-existing aerosol population, which typically suppresses nucleation and early growth. This creates an apparent paradox: despite a higher CS, NPF-P events exhibited significantly higher formation ($J_{2.5}$) and growth rates (GR). This indicates that the enhancement in precursor vapor concentrations (e.g., $H_2SO_4$, $HNO_3$, and likely organic vapors) under polluted conditions was sufficiently strong to overcome the inhibitory effect of the higher condensation sink. This scenario, where high vapor concentrations override a moderately elevated CS to drive intense NPF, has been documented in other polluted environments (Yang et al., 2021).

We have revised the manuscript to include the CS calculations and to integrate this crucial parameter into the mechanistic discussion.

First, the diurnal pattern of CS was added in Figure 2, and the following discussion was added in Section 3.2:

"To elucidate the factors driving distinct NPF behaviors, this section presents a diurnal comparison of key parameters between clean (NPF-C) and polluted (NPF-P) event days. As shown in Figure 2a, the average formation rate ($J_{2.5}$) during NPF-P events was 2.4 cm$^{-3}$ s$^{-1}$, approximately 3.6 times higher than during NPF-C events (0.7 cm$^{-3}$ s$^{-1}$). The peak $J_{2.5}$ in NPF-P events (6.2 cm$^{-3}$ s$^{-1}$ at 12:00 LT) was also higher and occurred one hour later than the peak in NPF-C events (1.8 cm$^{-3}$ s$^{-1}$ at 11:00 LT). The most pronounced enhancement—a fivefold increase—was observed at 10:00 LT (2.5 vs. 0.5 cm$^{-3}$ s$^{-1}$). While gaseous sulfuric acid concentrations were higher during NPF-P events, the condensation sink (CS) was also elevated (0.013 s$^{-1}$ vs. 0.008 s$^{-1}$ for NPF-C). Typically, a higher CS suppresses nucleation. The observed stronger formation and growth rates under these conditions therefore indicate that the enhanced production of condensable vapors from anthropogenic pollution was sufficient to overcome the increased sink strength, enabling intense NPF which has been documented in other polluted environments (Yang et al., 2021). Note that compared with the ~3.6-fold difference in $J_{2.5}$, the difference in gaseous sulfuric acid concentration between the two event types (23.2%) is insufficient to explain the magnitude of the difference in formation rate."

Second, we added discussion to address the interplay between elevated precursors and CS here.

"In addition, the chemical composition itself was shaped by the precursor environment. Although the condensation sink (CS) was elevated during NPF-P events (0.013 s$^{-1}$ vs. 0.008 s$^{-1}$ for NPF-C), which typically suppresses nucleation, significantly higher concentrations of gaseous sulfuric acid ($H_2SO_4$) and nitric acid ($HNO_3$) were present (Figure 2b, S3). This indicates that the enhanced production of condensable inorganic vapors under pollution transport was sufficient to overcome

the increased vapor sink, thereby promoting intense nucleation and growth."

**L370:** What does "TD" mean?

[*Response*] We thank the reviewer for asking for clarification on "TD". TD refers to the Thermal Denuder system used in tandem with the SMPS (TD-SMPS). We apologize for not defining this abbreviation in the main text. The method is described in detail in Section 2.1. The system heats the aerosol stream to 300 °C, allowing us to calculate the volume fraction remaining (VFR) after the evaporation of volatile and semi-volatile components. A higher VFR indicates a greater proportion of low-volatility or non-volatile material in the particles. To clarify it, we revised this part as follow:

"Support for this mechanism comes from Thermal Denuder (TD) measurements, which showed a higher volume fraction remaining (VFR) at 300 °C for NPF-P events (Figure 4d), indicating a greater proportion of low-volatility/non-volatile (refractory) material consistent with a processed, low-κ organic fraction."

**L368-369:** "HNO₃ enhances low-volatility organic compound production, further suppressing the hygroscopicity of NPF-P ultrafine particles" → It is not clear to me. How does the enhancement of organic compounds suppress the hygroscopicity when the κ consists of organic and inorganic substances? Please could you elaborate?

[*Response*] Thanks for reminding this. HNO₃ was suggested to play a dual role during the growth processes of the newly formed particles. First, it contributed directly to particle growth via the formation of ammonium nitrate. Second, as a strong oxidant, HNO₃ (often in conjunction with other oxidants like OH) enhances the atmospheric oxidation of volatile organic compounds (VOCs), promoting the formation of low-volatility oxygenated organic compounds (LV-OOCs). The condensation of these LV-OOCs increases the organic mass fraction, and organics generally have much lower hygroscopicity than sulfates or nitrates. Thus, enhanced HNO₃ influences particles via two pathways: directly contributing inorganic nitrate mass, and indirectly increasing the yield of low-volatility organics through chemical oxidation, thereby lowering the particle's overall average κ. We have now clarified it in the revised text.

"Notably, HNO₃ played a dual role. First, it contributed directly to particle growth via the formation of ammonium nitrate. Second, as a strong oxidant, HNO₃ (often in conjunction with other oxidants like OH) enhances the atmospheric oxidation of volatile organic compounds (VOCs), promoting the formation of low-volatility oxygenated organic compounds (LV-OOCs). The condensation of these LV-OOCs further increases the organic mass fraction of the growing particles. This pathway, where HNO₃ indirectly promotes the condensation of low-κ organic material, provides a chemical mechanism for the observed suppression of average particle hygroscopicity (κ) in NPF-P events."

**L373-374:** The authors write about figure 4b and then discuss figure 4d. This makes it difficult to follow the discussion.

[*Response*] Thank you for pointing out this. To make it clear, we have deleted this sentence.

**L374-375:** "During initial nucleation (0~2 hours), elevated non-volatile fractions (Figure 4d) suppress hygroscopic growth, maintaining $D_a$ at higher levels (~120 nm)." → This is difficult to follow. Where does this outcome stem from? It is quite vague.

[*Response*]We thank the reviewer for raising these specific concerns regarding clarity and supporting evidence. We admit that the link between elevated non-volatile fractions (VFR) and the maintenance of a high $D_a$ was not clearly explained. We have revised this statement to explicitly connect the observation (high VFR) to its physical implication (lower average particle hygroscopicity, κ) and the direct consequence for CCN activation (higher $D_a$). The revised text was below:

"The efficiency with which newly formed particles evolve into CCN is governed by the interplay between their dynamic growth and concurrent changes in hygroscopicity, as illustrated in Figure 4. During the initial hours of NPF events, particle volatility analysis reveals an elevated non-volatile fraction (high VFR; Figure 4d). This indicates a substantial presence of low-hygroscopicity material, such as highly oxidized organics, which lowers the effective particle hygroscopicity (κ). As a direct consequence, the critical activation diameter ($D_a$) peaks at ~124 nm for NPF-C and ~129 nm for NPF-P events in this phase (Figures 4a, S4a), since less-hygroscopic particles require a larger dry size to activate."

**L375-377:** Where does this outcome come from, and how is it depicted?

[*Response*] We thank the reviewer for raising these specific concerns. This conclusion is drawn from the time series data presented in Figure 4b, which shows the diurnal evolution of both NCN (blue line) and the activation ratio AR (red line). We now revised and clarify it.

"The diurnal evolution of the particle population further elucidates the transition from nucleation to CCN production. Total particle number concentration ($N_{CN}$) begins a rapid increase after ~07:00 LT, driven by the nucleation burst (Figure 4b). Although CCN concentration ($N_{CCN}$) starts to rise concurrently, the explosive production of small nucleation-mode particles initially causes the activation ratio (AR = $N_{CCN}/N_{CN}$) to decline, reflecting the time required for growth to CCN-active sizes. $N_{CCN}$ subsequently peaks around 09:00-10:00 LT, approximately 2-3 hours after the $N_{CN}$ surge, marking the period when a substantial fraction of new particles has grown sufficiently. After ~14:00 LT, as growth processes intensify (indicated by high GR), an increasing number of particles reach $D_a$, and the AR begins a gradual recovery (Figures S4b-c)."

**L377-380:** This makes it difficult to follow the discussion. I am unable to see all this information on the figures.

[*Response*] We thank the reviewer for raising these specific concerns regarding the clarity of the temporal evolution. To address these points, we have added a new figure (Figure S4) that explicitly shows the diurnal variation of the critical diameter ($D_a$), activation ratio (AR), CCN number concentration (NCCN), and total particle number concentration (NCN) for both NPF

event types. The corresponding analysis and discussion based on this figure have been incorporated into the revised MS, as detailed in our previous point-by-point response and the updated Section 3.4.

[Figure]

Figure S4: The diurnal variation of critical activation diameter ($D_a$), activation ratio (AR), the number of cloud condensation nuclei ($N_{CCN}$) and total particle number concentration ($N_{CN}$) in NPF-C and NPF-P events. The blue line denotes to NPF-C events and red line denotes to NPF-P events.

**L380:** Could you please add a tint to show the NPF days? It is difficult to examine the figures 4a, 4b, and 4c as they are.

[*Response*] We thank the reviewer for the suggestion. We have added shaded regions in Figure 4 to highlight the time periods during which NPF events occurred.

[Figure]

Figure 4: CCN-related parameters and chemical compositions across eight NPF events. (a) The solid line and the dashed line denote to the activation diameters at supersaturation (SS=0.2%) and supersaturation (SS=0.4%) during eight NPF events, respectively. (b) Temporal evolution of $N_{CCN}$ (blue solid line and blue dashed line) and its activation ratio (AR = $N_{CCN}/N_{CN}$, red solid and solid line). The solid line represents SS=0.2% and the dashed line represents SS=0.4%. (c) Time-resolved mass concentrations of particulate chemical constituents (organics, sulfate, nitrate, ammonium and chlorine) during the eight NPF events. (d) Solid line represents the fractional contribution of $H_2SO_4$ to GR within 2-20 nm particles; dashed line represents the non-volatile volume fraction remaining (1-VFR) in the 14-120 nm size bin. The blue line denotes to NPF-C events and blue line denotes to NPF-P events. (e-f) Diurnal variations in mass fraction contributions of chemical constituents during NPF-C and NPF-P events, respectively.

**L386-389:** What are the main differences when compared to Figures 2h and 2i?

[*Response*] We thank the reviewer for this clarification. Figures 2h and 2i show the mean diurnal profiles of the mass concentrations of non-refractory PM$_{2.5}$ chemical species (organics, sulfate, nitrate, ammonium, chloride) and black carbon (BC) for NPF-C and NPF-P event days, respectively. They illustrate how the absolute amount (in µg m$^{-3}$) of each component varies throughout the day for each event type. Figures 4e and 4f show the diurnal variations in the mass fraction contributions (i.e., the relative percentage) of the same chemical constituents for NPF-C and NPF-P events, respectively. These panels reveal how the relative composition of the aerosol changes over time, independent of the total mass loading. This is crucial for understanding which components dominate the particle phase during different stages.

**L390-393:** How is this outcome supported? It seems to be a general conclusion that lacks scientific argumentation.

[*Response*] We thank the reviewer for raising these specific concerns, We now provided more discussion in the revised version to clarify this issue.

"The VFR in the 14-80 nm size range was 10-20 % (Figure 4d), significantly higher than values reported for polluted urban Beijing (~5 %; Wu et al., 2017). Because heating to 300 °C effectively removes volatile inorganic salts and semi-volatile organic compounds, a higher VFR primarily reflects a greater abundance of low-volatility organic compounds (LVOCs). At our background site, where local combustion influence is minimal, this points to a more aged, oxidized organic aerosol component (Ehn et al., 2014; Jimenez et al., 2009), consistent with the observed lower $\kappa$ and higher $D_a$."

**L428-431:** I cannot understand what is being said. What and where is the role of nitrate?

[*Response*] We thank the reviewer for this question and apologize for the lack of clarity. The statement was intended to highlight a distinct diurnal pattern in the mass fraction contribution of nitrate between the two event types, as shown in Figures 4e and 4f. The role of nitrate, as a key condensable inorganic species, is to provide material for particle growth via the formation of ammonium nitrate. Our observation is that in NPF-P events (Figure 4f), the relative contribution of nitrate to the particle mass increased progressively in the afternoon and evening, particularly after ~15:00 LT. This suggests that nitrate formation became an increasingly important driver of particle growth as the day progressed under polluted conditions, likely due to sustained high levels of gaseous $HNO_3$ and $NH_3$. In contrast, in NPF-C events (Figure 4e), the nitrate mass fraction remained relatively stable and low throughout the day, indicating that nitrate played a minor role in growth compared to organics and sulfate. We have revised the text to make this distinction and the significance of the observation much clearer.

"The accelerated kinetics in NPF-P events can be attributed to the synergistic effects of elevated precursor concentrations and enhanced atmospheric oxidation. While transported oxidation products like highly oxygenated organic molecules (HOMs) may slightly suppress particle hygroscopicity, the concurrent surge in condensable inorganic vapors—particularly ammonium nitrate, as evidenced by the growing nitrate fraction in the afternoon and evening (Figures 4e-f)—provides a powerful and sustained driver for rapid condensational growth. Once partitioned into the particle phase, ammonium nitrate increases the overall particle hygroscopicity ($\kappa$). This physicochemical effect counteracts the hygroscopicity suppression by organics, effectively lowering the critical activation diameter ($D_a$) at a given supersaturation and facilitating the activation of growing particles into CCN. This combination of factors enables particles to overcome the initial hygroscopicity limitation and efficiently reach CCN sizes. In contrast, under cleaner conditions (NPF-C), the nitrate fraction remains low and stable (Figure 4e), signifying a minimal role in the growth process and leading to slower growth that extends the CCN conversion window."

***Section 3.4 should be completely revised. It is unclear and difficult to read. It needs to be improved scientifically.***

[*Response*] We sincerely thank the reviewer for the constructive feedback on Section 3.4. We acknowledge that the original section lacked clarity and scientific rigor, making it difficult to follow. We have undertaken a comprehensive, structural revision of Section 3.4 to address this concern. The section has now been completely restructured into three subsections, each with a clear focus:

3.4.1 Chemical Drivers of Varied Hygroscopicity and Critical Diameter:

This subsection explicitly links observed chemical composition (higher organic fraction, elevated VFR) to particle hygroscopicity ($\kappa$) and the resulting critical activation diameter ($D_a$), providing a mechanistic foundation.

3.4.2 Temporal Evolution of Particle Growth and CCN Activation Efficiency:

This part clearly describes the diurnal sequence from nucleation to CCN activation, using the data in Figure 4 to explain the time lag between $N_{CN}$ increase and $N_{CCN}$ peak, and the recovery of the activation ratio.

3.4.3 Quantitative Assessment of NPF-to-CCN Conversion Efficiency and Kinetics:

This final subsection introduces and integrates the two key metrics—$EF_{CCN}$ and the new "Time Window ($\tau$)"—to quantitatively compare the efficiency and speed of CCN production between event types, and validates the $\tau$ concept with external data.

We believe this thorough revision has significantly improved the clarity, scientific depth, and readability of the section.

**References**

[revised manuscript text omitted]